

# Stratospheric impact on the Northern Hemisphere winter and spring ozone interannual variability in the troposphere

Junhua Liu[1,2], Jose M. Rodriguez[2], Luke D. Oman[2], Anne R., Douglass[2], Mark A. Olsen[3,4], Lu Hu[5]

[1] Universities Space Research Association (USRA), GESTAR, Columbia, MD, USA
   [2] NASA Goddard Space Flight Center, Greenbelt, MD, USA
   [3] TriVector Services Inc., Huntsville, AL, USA
   [4] NOAA/OAR/Office of Weather and Air Quality
   [5] Department of Chemistry and Biochemistry, University of Montana, Missoula, MT, USA

*Correspondence to*: Junhua Liu (junhua.liu@nasa.gov)

**Abstract.** In this study we use $O_3$ and stratospheric $O_3$ tracer simulations from the high-resolution Goddard Earth Observing System, Version 5 (GEOS-5) Replay run (MERRA-2 GMI at 0.5° model resolution ~50 km) and observations from
ozonesondes to investigate the interannual variation and vertical extent of the stratospheric ozone impact on tropospheric ozone. Our work focuses on the winter and spring seasons over North America and Europe. The model reproduces the observed interannual variation of tropospheric $O_3$, except for the Pinatubo period from 1991 to 1995 over the region of North America. Ozonesonde data show a negative ozone anomaly in 1992-1994 following the Pinatubo eruption, with recovery thereafter. The simulated anomaly is only half the magnitude of that observed. Our analysis suggests that the simulated
Stratosphere-troposphere exchange (STE) flux deduced from the analysis might be too strong over the North American (50°N-70°N) region after the Mt. Pinatubo eruption in the early 1990s, masking the impact of lower stratospheric $O_3$ concentration on tropospheric $O_3$. European ozonesonde measurements show a similar but weaker $O_3$ depletion after the Mt. Pinatubo eruption, which is fully reproduced by the model. Analysis based on a stratospheric $O_3$ tracer (StratO3) identifies differences in strength and vertical extent of stratospheric ozone influence on the tropospheric ozone interannual variation
(IAV) between North America and Europe. Over North America, the StratO3 IAV has a significant impact on tropospheric $O_3$ from the upper to lower troposphere and explains about 60% and 66% of simulated $O_3$ IAV at 400 hPa, ~11% and 34% at 700 hPa in winter and spring respectively. Over Europe, the influence is limited to the middle to upper troposphere, and becomes much smaller at 700 hPa. The stronger and deeper stratospheric contributions in the tropospheric $O_3$ IAV over North America shown by the model is likely related to ozonesondes' being closer to the polar vortex in winter with lower
geopotential height, lower tropopause height, and stronger coupling to the Arctic Oscillation in the lower troposphere (LT) than over Europe.



# 1 Introduction

Tropospheric ozone plays an important role in the oxidative chemistry of the troposphere. It is the third most important climate forcing gas after carbon dioxide and methane, and affects the radiative balance of the atmosphere (Forster et al., 2007). Unlike the well-mixed greenhouse gases, tropospheric ozone and its radiative forcing are spatially and temporally inhomogeneous (Lacis et al., 1990; Forster and Shine, 1997; Joiner et al., 2009; Worden et al., 2008; 2011; Bowman et al., 2013). Stratosphere-troposphere exchange (STE) has been shown to impact the tropospheric ozone distribution (e.g., Terao et al., 2008; Hess et al., 2015; Holton et al., 1995). Liu et al. (2017) showed that stratospheric input plays a dominant role in driving the interannual variation (IAV) of upper tropospheric ozone over the southern hemisphere ocean, where its radiative impact is largest. Considering the observed and expected net global decrease in emissions of ozone precursors and the predicted increase in ozone STE (e.g., Collins et al., 2003; Sudo et al., 2003; Hardiman et al., 2014; Banerjee et al., 2016), it is important to quantify the role of STE compared to that of precursor emissions in determining tropospheric ozone variations.

In this study we use a long-term Goddard Earth Observing System (GEOS) - chemistry climate model (CCM) replay simulation of $O_3$ and 'stratospheric ozone tracer diagnostic' (StratO$_3$), as well as the model's meteorological fields, to interpret the IAV obtained from the ozonesonde records in the northern hemisphere troposphere at mid-high latitudes. In so doing, we examine the vertical and longitudinal distribution of the stratospheric $O_3$ impact on the IAV of tropospheric $O_3$ and their linkage to dynamical parameters.

STE has been widely studied for several decades (Danielsen, 1968; Holton et al., 1995; Olsen et al., 2002; 2003; 2013; Stohl et al., 2003a; 2003b; Sprenger and Wernli, 2003; Thompson et al., 2007; Lefohn et al., 2011; Skerlak et al., 2014). It contributes significantly to ozone in the upper troposphere, where ozone has a strong radiative effect. Observations, assimilations and simulations from high resolution models show that deep STE events occasionally reach ground level, adversely affecting the air quality near the surface (e.g., Haagenson et al., 1981; Davies and Schuepbach, 1994; Lefohn et al., 2001; Lin et al., 2012; 2014; Ott et al., 2016; Knowland et al., 2017; Akritidis et al., 2018). In addition, various chemistry climate models project increased STE leading to a higher contribution of stratospheric ozone to tropospheric ozone (Collins et al., 2003; Sudo et al., 2003; SPARC-CCMVal, 2010; Zeng et al., 2010). Limitations in the representation of subscale processes lead to large uncertainties in the calculated stratospheric contribution to concentrations and variations of tropospheric ozone. These limitations also lead to uncertainty in their relative magnitudes compared to the effects of increased or decreased emissions of ozone precursors. The uncertainties in stratospheric contribution to tropospheric ozone variations lead to similar uncertainties in resulting ozone radiative forcing, a key area of focus in climate change studies.

Various studies have used tropospheric chemistry transport models (CTMs) to examine the response of tropospheric ozone to changes in stratospheric input and in surface emissions; these models used a simple treatment of stratospheric-tropospheric flux, adopting either the SYNOZ (synthetic ozone) approximation developed by McLinden et al. (2000) to specify the stratosphere-to-troposphere flux (e.g., the GEOS-Chem model in Hess and Zbinden, 2013; Fusco and Logan, 2003), or





specifying ozone in the lower stratosphere (LS) (the GISS model in Fusco and Logan, 2003; Karlsdottir et al., 2000). Hess et al. (2015) analyzed the effects of stratospheric input to tropospheric ozone variations over the northern hemisphere mid-latitudes with four ensemble simulations of the free running Whole Atmosphere Community Climate Model (WACCM) for 1953 to 2005. Their model used a standard stratospheric chemical mechanism and simple $CH_4$-$NO_X$ chemistry in the troposphere with constant surface emissions of ozone precursors. The study reproduced well the observed tropospheric $O_3$

IAV, suggesting that natural variability in transport and stratospheric ozone plays a significant role in the tropospheric ozone IAV over the northern hemisphere.

In this study, we use a long-term full chemistry $O_3$ simulation and an online stratospheric $O_3$ tracer simulation with horizontal resolution of 0.5°, suggested to be the minimum model resolution needed to resolve the structure of deep STE events (Ott et al., 2016). We focus on 1990 - 2016, a period of considerable IAV in STE (James et al., 2003), varied trends in

emissions of ozone precursors, and greater availability of reliable ozone observations than in prior periods. We examine the vertical extents of STE impact on tropospheric ozone using model simulations and ozonesonde measurements sampled over North America and Europe. We rely on the StratO$_3$ tracer simulation to quantify the contribution of stratospheric $O_3$ to tropospheric $O_3$ at different levels, as well as its contribution to the IAV.

## 2 Data and Model

### 2.1 Ozonesondes

We select 17 ozonesonde sites including eight from North America and nine from Europe, all of which have a record of at least 3 profiles every month between 1990 and 2016 (Figure 1 and Tables 1 & 2). The data are obtained from the World Ozone and Ultraviolet Data Center (WOUDC, http://www.woudc.org). Observations over most stations were obtained using electrochemical concentration cell (ECC) ozonesondes, which rely on the oxidation reaction of ozone with potassium iodide

in solution (Komhyr et al., 1995). At Hohenpeissenberg, Germany, observations were obtained using the Brewer/Mast ozonesonde. The sondes profiles have a vertical resolution of ~150 m for ozone, with an accuracy about ±5% in the troposphere (WMO, 2014).

### 2.2 MERRA2-GMI

We use a replay simulation (Orbe et al., 2017) of the GEOSCCM with the Global Modeling Initiative (GMI) chemical

mechanism (Strahan et al., 2007; Duncan et al., 2007) for trace gas chemistry, which includes a complete treatment of stratospheric and tropospheric chemistry, and the Goddard Chemistry Aerosol Radiation and Transport (GOCART) module (Chin et al., 2002; Colarco et al., 2010) for aerosols. The replay simulation ingests essential output of MERRA-2 meteorology (U, V, T, pressure) every 6 h, and then utilizes the model physics and chemistry to calculate the evolution of meteorology and chemical composition from the ingested analysis and the species concentrations at the end of the previous

replay interval at every model time step over the 6 h replay interval. The replay simulation is run at a MERRA-2 native



resolution of ~50 km in the horizontal dimension and 72 vertical levels (http://acd-ext.gsfc.nasa.gov/Projects/GEOSCCM/MERRA2GMI). This replay simulation is referred to as the 'MERRA2-GMI" simulation.

The MERRA2-GMI simulation was run from 1980 to present. The emissions in this run include anthropogenic, biofuel,
biomass burning, and biogenic emissions. The values for fossil fuel and biofuel emissions are from the MACCity inventory (Granier et al., 2011) until 2010, and then derived by following the Representative Concentration Pathway (RCP) 8.5 scenario after 2010. The MACCity anthropogenic emissions are derived by interpolating the Atmospheric Chemistry and Climate - Model Intercomparison Project (ACCMIP) emissions (Lamarque et al., 2010) on a yearly basis between the base years 1990, 2000, 2005 and 2010. For the years 2005 and 2010, the interpolation follows the RCP 8.5 emission scenario.
Biomass burning emissions are from the Global Fire Emissions Dataset (GFED) version 4s (Giglio et al., 2013) after 1997. Prior to 1997, biomass burning emissions are based on a GFED4s climatology with year-to-year variability imposed using regional scale factors derived from the Total Ozone Mapping Spectrometer (TOMS) aerosol index (Duncan et al., 2003). The simulation used the Model of Emissions of Gases and Aerosols from Nature (MEGAN) (Guenther et al., 2006) to simulate biogenic emissions, including isoprene, within the model.  The lightning parameterization in the model (Allen et al. 2010) is
constrained by the MERRA-2 detrended cumulative mass flux, with seasonal constraints from the Lightning Imaging Sensor (LIS) / Optical Transient Detector (OTD) v2.3 climatology (Cecil et al., 2014). Methane is specified using latitude and time dependent surface observations from the NOAA Earth System Research Laboratory (ESRL) Global Monitoring Division (GMD) network (Dlugokencky et al., 2011).

A StratO$_3$ tracer is included in the model to simulate O$_3$ of stratospheric origin in the troposphere at all locations and times.
The StratO$_3$ tracer is defined relative to a dynamically varying tropopause tracer (e90) (Prather et al., 2011). The e90 tracer is set to a uniform mixing ratio (100 ppb) at the surface with a 90-day e-folding lifetime. In our simulations, the e90 tropopause value is set to 90 ppb. StratO$_3$ is set equal to O$_3$ in the stratosphere and is removed in the troposphere using chemical loss and deposition output from the standard full chemistry simulation. The MERRA2-GMI simulation has hourly output for ozone and three-hourly output for StratO$_3$ tracer at each model level. When comparing to the ozonesonde measurements, the model
outputs are sampled at the nearest grid point and launch time for each sonde.

## 3 Model evaluation with satellite observations

We first evaluate model performance by comparing the simulated total column O$_3$ with the version 8.6 merged total ozone datasets from the Solar Backscatter Ultraviolet (SBUV)  (McPeters et al., 2013; Frith et al., 2014). The SBUV observing system measures concentrations at different levels from the ground to the top of the atmosphere, with a vertical resolution
changing from 6 km resolution in the middle and upper stratosphere to about 15 km in the troposphere. The SBUV v8.6 merged total ozone dataset are monthly-mean zonal and gridded average products from 1970 to 2017 constructed by combining individual data sets of ozone from a series of SBUV instruments – the Nimbus 4 BUV, the Nimbus 7 SBUV, and



SBUV/2 instruments (https://acd-ext.gsfc.nasa.gov/Data_services/merged/). Figure 2 compares the zonal mean of the simulated and observed total ozone columns (TOZ) averaged over 30°N-60°N and 60°N-75°N as well as their anomalies

from 1991 to 2017. The anomalies are calculated by removing the monthly mean averaged from 1991 to 2017. The model reproduces the magnitude and seasonal cycle of the observed total column ozone over the mid-high latitudes of the northern hemisphere. Although the model has a low bias compared to the observations, the discrepancy is not statistically different from zero. The model reproduces the observed IAV well, showing more positive ozone anomalies in 2000s and negative ozone anomalies in early 1990s.

We then compare the tropospheric $O_3$ column between model and values derived from a combination of measurements from the Aura Ozone Monitoring Instrument (OMI) and Microwave Limb Sounder (MLS) for January 2005 to December 2016 (Ziemke et al., 2019). The OMI/MLS tropospheric ozone column (TCO) is determined by subtracting the MLS stratospheric ozone column (SCO) from OMI total column ozone each day at each grid point from 60°S to 60°N. The tropopause pressure is defined using the World Meteorological Organization (WMO) 2K-km$^{-1}$ lapse-rate definition from the National Centers for

Environmental Prediction (NCEP) re-analyses. The data set has included a +2 DU offset correction and a -0.5 DU/decade drift correction following evaluation with ozonesondes, cloud slicing measurements, and the OMI row anomaly. More detailed description of this dataset is given in Ziemke et al. (2019). We select the same definition of the tropopause pressure for the model simulation to calculate tropospheric column ozone in the model.

Figure 3 shows the tropospheric ozone columns and their anomalies from the MERRA2-GMI replay simulation, together

with the OMI/MLS TCO averaged between 30°N and 60°N. The anomalies are calculated relative to their respective 2005 to 2017 monthly mean. The MERRA2-GMI simulation reproduces well the phase of observed seasonal cycles, but underestimates slightly the observed summer maxima (Figure 3a). The correlation between the OMI/MLS TCO and the MERRA2-GMI TCO decreases from 0.93 for the ozone concentrations to 0.67 for the anomalies, but in general model and observations show similar magnitude, IAV and trend, with more negative anomalies before 2009, followed by a continuous

increase. Thus, the MERRA2-GMI replay simulation results are in good agreement with the seasonality and IAV of the total and tropospheric column ozone from satellite observations.

## 4: Results

### 4.1 $O_3$ IAV for all sondes and seasons.

The monthly means of observed (black) and simulated (red) $O_3$ anomalies averaged over the selected seventeen ozonesonde

sites at 200 hPa, 400 hPa and 700 hPa are shown in Figure 4. The shaded area represents the 95% confidence interval (CI) of the observed mean. The simulated anomalies are calculated from the output at the grid point nearest the ozonesonde site on the date of the measurement. Anomalies are smoothed with a 6-month running mean to highlight the interannual variations. In the lower stratosphere (200 hPa), both the phase and magnitude of the simulated anomalies agree very well with that derived from observations (r = 0.93). For example, both observation and simulation show a strong negative $O_3$ anomaly of



about 166 ppb with ~39% relative change in February 1993 after the Pinatubo eruption. Both show strong positive anomalies
in the winter-spring seasons of 1991, 2013, and some years after strong El Ninos, e.g., 1998-1999, 2001-2004, 2009-2010,
with an increased stratospheric circulation (e.g., Neu et al., 2014).

In the troposphere, the model is in reasonable agreement with the phase of observed anomalies, but tends to underestimate
the magnitude. For example, the model underestimates the observed $O_3$ depletion after the Mt. Pinatubo eruption at 400 hPa

(relative anomalies of -14% in measurements vs. -8% in model) and 700 hPa (-13% vs. -5%), as well as the observed
maximum $O_3$ increase in February 2013 at 400 hPa, and in December 2003 at 700 hPa. Still, most of model results fall within
the 95% CI of observations.

Both observations and simulations show the largest interannual variations in the winter and spring, when the strongest IAVs
occur. In the next section, we examine time series for these two individual seasons over North America and Europe and

investigate their controlling factors.

### 4.2 Winter and spring $O_3$ IAV in the lower stratosphere and troposphere over North American and European sites

Figure 5 compares the anomalies of modeled and observed ozone at 200 hPa, 400 hPa and 700 hPa in the winter and spring
seasons from 1990 to 2016 averaged over sites from North America and Europe. Anomalies at each site are calculated by
removing the respective seasonal mean climatology from 1995 to 2016, and then averaged over all sites for each region. The

shaded area represents the 95% CI of the calculated mean from daily observations over all the selected stations.

At 200 hPa, the model reproduces very well the observed IAV in both seasons over both regions ($r \geq 0.91$). The IAV of $O_3$ is
larger over North America than over Europe, and larger in spring than in winter. Negative $O_3$ anomalies occur in the early
1990s and at the end of the record from 2014 to 2016, while positive anomalies are obtained for most years between 1998
and 2013. The negative ozone anomalies during the period of 1992-1996 are consistent with the chemical and dynamical

perturbations following the June 15th 1991 eruption of Mt. Pinatubo  (Hadjinicolaou et al., 1997; Rozanov et al., 2002;
Stenchikov et al., 2002). The negative ozone anomaly in 2015-2016 is associated with stratospheric circulation changes
caused by the unusually warm ENSO event aligned with a disrupted Quasi-Biennial Oscillation (QBO) during that period
(Tweedy et al., 2017; Diallo et al., 2018)

Both observed and simulated $O_3$ IAV at 200 hPa are correlated with the temperature at 150 hPa averaged over latitudes north

of 60°N ($r > 0.7$ over North America and $r \geq 0.55$ over Europe, Figure S1) in the winter. The correlation is weaker in the
spring. These results are consistent with our understanding of the impact of temperature variations on the formation of polar
stratospheric clouds and polar vortex isolation with reduced transport of $O_3$ from the tropics at low temperatures (e.g.,
Schoeberl and Hartmann, 1991), all of which modulate ozone concentrations in the lower stratosphere during winter/spring.

The IAV at 400 hPa over North American sites is similar to those at 200 hPa (Figure 5 e-f). The model reproduces the

overall variations as inferred from observations, showing negative anomalies in early 1990s, with mostly positive anomalies
thereafter. The observed $O_3$ depletion after the Mt. Pinatubo eruption reaches its maximum amplitude of 7 ppb (-13%
relative anomaly) in the winter 1992-1994. The model underestimates the observed peak depletion in winter of 1992, with



the simulations falling outside the 95% CI of the observations from 1992 to 1994 (Figure 5 e).  In spring, the model reproduces well the timing of observed O₃ depletion, but again underestimates its amplitude (Figure 5 f). Over European sites, the observed O₃ IAV, excluding year 1990-1991, exhibits a similar pattern to the one at 200 hPa after 1991, although the minima after the Mt. Pinatubo eruption are not as pronounced as over North America. The maximum positive anomaly, observed in 1990-1991, is not reproduced by the model. The model-observation correlation coefficients increase significantly if we omit these two years (from 0.18 to 0.58 in the winter and from 0.43 to 0.58 in the spring).

At 700 hPa, the observations from the North American sites show a similar negative ozone anomaly in 1992-1994 to that obtained at 200 hPa and 400 hPa, with prevailing positive anomalies thereafter. The model results for the phase of the interannual variations are in relatively good agreement with observations, but again underestimate the magnitudes of the negative anomalies in the early 1990s after the Mt. Pinatubo eruption. Over the European sites, unlike that over North America, the model reproduces the amplitude of observed O₃ depletion after the Mt. Pinatubo eruption and shows similar magnitude of observed variations.

Table 3 shows the correlation coefficients of the winter and spring ozone IAVs between 200 hPa and 400 hPa, 200 hPa and 700 hPa for the observations and simulations averaged over the North American and European stations. Both the model and observations suggest that about 27% of North American interannual ozone variability at 400 hPa is related to changes at 200 hPa in the winter. The explained variance is higher in the spring with 40% and 46% respectively in the observation and simulation. Over Europe, the 200-400 hPa O₃ correlation in the observation is relatively low (r=0.31 in DJF and 0.13 in MAM), due to the phase shift of these two-time series during the first two years, where observed O₃ anomalies are negative at 200 hPa, but reach a maximum at 400 hPa. The correlation increases to 0.67 after removing these two years in the winter, but not that much in the spring (r= 0.23).  High correlations of the O₃ IAV between 200 hPa and 400 hPa is seen in the model in both seasons.  The highest correlation coefficient between 200 hPa and 700 hPa is found over the North American sites in the spring with r = 0.46 & 0.41 respectively in observations and simulations, which is consistent with the previous findings of the deep STE hot spots along western U.S. in the spring season (Skerlak et al., 2014).

The correlation analysis between the stratosphere and troposphere IAV in both observations and model simulations suggest a potential impact of stratospheric O₃ on tropospheric O₃ variations. Previous studies have found high correlations between ozone in the lower stratosphere with that in the middle and lower troposphere with the largest effects in late winter and spring. Correlation does not necessarily mean causality, and to date model investigations of this correlation(Terao et al., 2008; Hess and Zbinden, 2013) did not use a model with both stratospheric and tropospheric chemistry, and realistic stratospheric circulation.  The MERRA2-GMI simulation has both of these attributes, detailed dynamic diagnostics, and the StratO₃ tracer as described in Section 2.2. In the next section, we use the StratO₃ tracer to examine the contribution of stratospheric ozone to the IAV of tropospheric ozone, as a function of altitude, season, and location. We will also use diagnostics from the model to explore the influence of dynamics on the stratospheric O₃ contribution to the tropospheric O₃ and its IAV.



**4.3: Impact of stratospheric O₃ on tropospheric O₃ IAV**

Figure 6 shows the same comparison between the observed (black lines) and simulated ozone (red lines) anomalies as in Figure 5, but adding the anomalies of simulated StratO₃ tracer (green lines). As expected, the StratO₃ tracer anomalies at 200 hPa are almost identical to the simulated O₃ anomalies, since most measurements are in the stratosphere at this level.

The IAV of StratO₃ tracer in the troposphere reflects a combined effect of the changes in the lower stratospheric O₃ concentrations and in the strength of stratosphere-to- troposphere (STE) mass flux. These two effects may either cancel or reinforce each other, depending on their relative phases. At 400 hPa, over the North American stations, the minimum and maximum of StratO₃ tracer is highly correlated with the minimum and maximum of simulated O₃. The IAV of StratO₃ explains more than 60% of simulated O₃ variations (r = 0.77 in DJF and 0.81 in MAM), suggesting that the changes of

stratospheric O₃ input provides a significant influence on the simulated O₃ IAV in the upper troposphere. The correlation between StratO₃ and observed O₃ is slightly lower (0.44) than that with simulated O₃ in DJF over North America. The decreased correlation is mainly due to the model-observation discrepancy between 1992-1994. The sondes at 400 hPa show a similar O₃ depletion through 1992-1994 as seen at 200 hPa after the Mt. Pinatubo eruption, while the model shows an O₃ increase after 1992 through 1994, which is driven by changes in the stratospheric O₃ contribution to the modeled O₃ (Figure

6e). This suggests that the impact of the negative anomalies of stratospheric ozone (200 hPa) may be counterbalanced by an increase in downward mass flux from the stratosphere, thus leading to the model underestimation of the negative anomaly in observations at 400 hPa. In MAM, the StratO₃-O₃ correlation to the observed O₃ stays high (0.74) over North America. Over European sites, a similar correlation is observed between simulated O₃ and StratO₃ at 400 hPa in the winter (r = 0.78), with a slightly smaller value in the spring (r=0.61). The correlation decreases when comparing StratO₃ to the observed O₃, mainly

because of the model-observation discrepancy during the first two years. Omitting the first two year gives a fair correlation between StratO₃ and observed O₃ (0.66 in DJF and 0.34 in MAM). The fair to good correlations between StratO₃ and observed O₃ give credence to the reality of the impact of stratospheric ozone on the troposphere. In general, the good agreement between ozone IAV with that of StratO₃ at 400 hPa indicates that changes in the stratospheric ozone contribution play an important role in the simulated upper tropospheric O₃ IAV in winter-spring over North America and Europe.

The bottom panel of Figure 6 compares the simulated StratO₃ tracer anomalies to the observed and simulated O₃ anomalies at 700 hPa over North American and European ozonesonde sites in the winter and spring seasons. Over North America, the observed O₃ anomalies stay low in the early 1990s, and increase thereafter in both seasons, which is underestimated in the model. In the winter, StratO₃ anomalies decrease slightly in contrast to increases in both observed and simulated O₃ anomalies. The winter StratO₃-O₃ correlation is ~ 0. In spring, sonde and model exhibit similar IAV of O₃ and are similar to

the phase of the IAV of the StratO₃ after the Pinatubo period. The StratO₃-O₃ correlation increases from 0.07 to 0.33 in winter and from 0.36 to 0.58 in spring after omitting the Pinatubo period.

Over North America, our model results are in good agreement with the observed IAV at all levels except right after the Mt. Pinatubo eruption. The model only reproduces about half of the observed tropospheric depletion over North America. As





discussed above, this is likely due to an excessive mass flux from the stratosphere in the MERRA-2 analysis during this

period. Model results are in better agreement with the magnitude of observed $O_3$ depletion after the Mt. Pinatubo eruption in the middle and lower troposphere over Europe. We also note that the anomalies in $StratO_3$ diverge from those of simulated $O_3$ towards the end of our analysis period, particularly at 700 hPa. This is to be expected since the impact of stratospheric ozone decreases at lower altitudes, and the impact of ozone production from its precursors becomes more important. In summary, our model analysis identifies differences in the strength and vertical extent of stratospheric ozone impact on the

tropospheric ozone IAV between North America and Europe. Over North America, the $StratO_3$ IAV has a significant impact on the tropospheric $O_3$ IAV from the upper to lower troposphere and explains about 60% and 66% of the simulated $O_3$ IAV at 400 hPa, ~11% and 34% at 700 hPa in winter and spring respectively. Over Europe, the influence is limited to the middle to upper troposphere, and becomes much less at 700 hPa. The difference in the stratospheric $O_3$ influence between North America and Europe is likely due to the longitudinal difference in dynamics.

Previous studies have suggested that the IAV of the STE mass flux is likely correlated to changes in the tropopause height (e.g., Gettelman et al., 2011). The top panel of Figure 7 shows the comparison of the observed $O_3$ mixing ratio anomalies at 400 hPa and the tropopause pressures derived from the observed $O_3$ profiles following the criteria in vertical gradient and $O_3$ mixing ratio given by Browell et al. (1996). The tropopause pressure was estimated to be at the pressure where a linear regression line passing through the lower stratospheric $O_3$ profile (150 ppb - 400 ppb, lower than 100 hPa) intersects with the

100 ppb $O_3$ level. The bottom panel of Figure 7 compares the simulated $O_3$ and $StratO_3$ anomalies at 400 hPa with the tropopause pressures derived from simulated $O_3$ profiles following the same criteria as for the observations. As expected, the IAV of $O_3$ and $StratO_3$ positively correlates with that of the derived tropopause pressure (anticorrelates with the tropopause height) in both model and observation. In general, during years with a lower tropopause, stratospheric $O_3$ influence at 400 hPa increases and results in a positive $O_3$ anomaly. During years with a higher tropopause, decreased stratospheric $O_3$

influence leads to a negative $O_3$ anomaly at 400 hPa.

The above high correlations between the IAV of tropopause pressure and $StratO_3$ raise the question of what dynamical conditions control the higher/lower tropopause pressures, STE mass fluxes, and the subsequent impact of stratospheric ozone on tropospheric ozone. These questions are particularly important if these dynamical conditions may exhibit future changes as a result of climate change. In the next section, we rely on the model's 3-d dynamical diagnostics, including air mass flux

and horizontal wind patterns, to examine both the vertical and horizontal transport influence of the stratospheric $O_3$ contribution on the tropospheric $O_3$ and its IAV. We also examine the longitudinal difference in the model's dynamics to explain the identified longitudinal difference in stratospheric $O_3$ influence in the troposphere between North America and Europe.



## 5 Influence of dynamics

### 5.1 Case study of 3-d dynamic characteristics

The planetary-scale Rossby waves, including quasi-stationary Rossby waves and Rossby wave-breaking events, superimposed on the mean westerly zonal flow are the dominant dynamical variability over northern midlatitudes in winter and spring. Homeyer and Bowman (2013) have shown that Rossby wave-breaking occurring in the upper troposphere can affect the flow at all tropospheric levels and plays an important role in the meridional transport of both tropical and 295 subtropical air masses. Ozone transport associated with these wave disturbances are responsible for a large fraction of ozone temporal and spatial variability in winter and spring (e.g., Kinnersley and Tung, 1998; McCormack et al., 1998; Lozitsky et al., 2011; Zhang et al., 2015). Thorncroft et al. (1993) classified Rossby wave-breaking events as either "equatorward breaking" or "poleward breaking". In the equatorward breaking, tongues of high PV, stratospheric air extend equatorward associated with frequent STE processes, while, in the latter case, tongues of low PV, upper tropospheric air extend poleward. 300 In this section, we rely on the model's 3-d dynamic diagnostics, including air mass flux and horizontal wind patterns, to examine both vertical and horizontal transport influence of the stratospheric $O_3$ contribution to the tropospheric $O_3$ and its IAV. By doing that, we examine the linkages of the dynamical structures at the lower stratosphere to the stratospheric $O_3$ contributions in the upper and middle troposphere and how they vary with the changes in wave disturbances year by year. Our analysis first focuses on the year 1993, when there is a major discrepancy with the observations at 400 hPa as shown in 305 Figure 6.

Figure 8 illustrates the corresponding changes of $StratO_3/O_3$ ratio at 400 hPa to changes in dynamics including horizontal winds at 400 hPa, and vertical airmass flux near the seasonal mean tropopause pressure in the year 1993. The tropopause pressure in the model averaged from 30°N to 80°N is around 250 hPa in winter and around 300 hPa in spring. The top row of Figure 8 shows maps of simulated $StratO_3/O_3$ ratio in the winter and spring of 1993; prevailing wind patterns at 400 hPa are 310 superimposed on this ratio. The jet locations, approximated by the strongest winds, are indicated by red asterisks in the top row of Figure 8. The second row of Figure 8 shows the respective anomalies of simulated $StratO_3/O_3$. The third row shows the airmass flux around the tropopause pressure with 250 hPa in winter and 300 hPa in spring (Blue color represents the downward motion and red color represents the upward motion near the tropopause) and the fourth row shows the anomalies of airmass flux at the same pressure (Blue color represents an increase of downward flux or a decrease of upward flux. Red 315 color represents a decrease of downward flux or an increase of upward flux around the tropopause).

$StratO_3/O_3$ ratio represents the impact of stratospheric air on tropospheric ozone at this level. Regions with the maximum $StratO_3/O_3$ ratio at 400 hPa in general show a similar longitudinal distribution in the winter and spring seasons with a southward shift over eastern North America and a poleward shift over western North America and Europe. However, there are year by year variations in this longitudinal distribution of the $StratO_3/O_3$ ratios, associated with the IAV of wave 320 disturbances in the westerlies.





In the winter of 1993, strong northwesterly winds prevailed north of 50°N and the westerlies dominated between 30°N and 50°N over western North America. The winds converged around 45°N over eastern North America and moved northwestward into the North Atlantic and Europe. The winds changed direction to northwesterly over Europe, bringing higher stratospheric $O_3$ air into eastern Europe. The maps of the airmass flux and its anomalies suggest that North America

between 50°N and 70°N was dominated by more vigorous downward mass fluxes of stratospheric air. Meanwhile, the southeasterly winds brought ozone rich air from high latitudes, resulting in a positive anomaly of stratospheric ozone influence at 400 hPa (Figure 8a, c, d, f). Our results suggest that although the stratospheric $O_3$ depletion modulated this process, the enhanced STE in the model counteracted the depletion and reduced the negative anomalies expected at 400 hPa over North American between 50°N and 70°N (Figure S2). Over the high latitudes (> 70°N), where there is less dynamic

perturbation (including both vertical and horizontal transport), the stratospheric $O_3$ contribution at 400 hPa was largely driven by the depletion of the $O_3$ concentration in the lower stratosphere and showed a strong negative anomaly in 1993. Although most of the European region was covered by the increased downward airmass flux at 200 hPa in the winter of 1993, a negative anomaly of the StratO$_3$ contribution at 400 hPa was seen over western Europe. It is likely that the combined negative effects of the $O_3$ depletion in the lower stratosphere and the downwind of the low stratO$_3$ air from the subtropical

North Atlantic Ocean exceeded the positive effect of the increased downward airmass flux over this region.

In the spring of 1993, southwesterly wind prevailed south of 65°N over western North America, bringing in low StratO$_3$ oceanic air from the subtropics. The wind was deflected to the south around 120°W and 65°N and flowed to Hudson Bay around 60°N, then transported to the east until reaching the west coast of Europe, where the winds bifurcated into two branches: one passed by the northern side of Europe and the other flowed around the southern side of Europe. In North

America south of 50°N, there were three cells with increased upward airmass fluxes ranging from 110°W to 50°W, resulting in a decreased StratO$_3$ contribution in the downwind regions (Figure 8d, f, h). Most regions of western North America north of 50°N showed a decreased StratO$_3$ contribution, likely contributed jointly by the lower stratospheric $O_3$ and the decreased STE flux. Decreased stratospheric $O_3$ contribution occurred over most of Europe, especially the northwest coast, which was downwind of the westerly flows with low stratospheric $O_3$.

Figure 9 shows the similar analysis as Figure 8, except for 1998, when stratospheric $O_3$ levels have recovered from the Mt. Pinatubo eruption and reached a regional maximum (Figure 4). In the winter of 1998, a poleward shift of the jet occurred over most of North America. The jet location as well as the regions with the maximum StratO$_3$/$O_3$ ratio moved to the north by about 7° compared to the winter of 1993. Strong southwesterly winds combined with increased ascending air dominated over western North America between 45°N and 70°N and brought in tropical oceanic low $O_3$ air. Over middle and eastern

North America, weakened descending air resulted in a minimum of stratospheric $O_3$ influence over these regions. Therefore, although there was an increase in the stratospheric $O_3$ concentrations, the weaker STE flux associated with the northward movement of the jet system over North America produced only a small $O_3$ variation at 400 hPa.

Our analysis suggests that the IAV of wave disturbances in the westerlies likely affect the IAV of the regional distributions of prevailing wind patterns as well as the strength of STE flux. The IAV of stratospheric $O_3$ influence in the troposphere



reflects a combined effect of the changes in the lower stratospheric $O_3$ concentration and in the 3-d dynamics, which may either cancel or reinforce each other.

## 5.2 Longitudinal difference of stratospheric $O_3$ influence

Our model analysis identified differences in the strength and vertical extent of stratospheric ozone impact on tropospheric ozone IAV between North America and Europe. Over North America, the $StratO_3$ IAV has a significant impact on

tropospheric $O_3$ from the upper to lower troposphere. Over Europe, the influence is limited to the middle to upper troposphere. The difference of stratospheric $O_3$ impact between North America and Europe is caused by the longitudinal differences in dynamics.

Figure 10 and 11 show the latitudinal average (30°N to 80°N) of tropopause pressure, geopotential height at 400 hPa, and the $StratO_3/O_3$ ratio at 400 hPa at each longitude from 180°W to 180°E from 1990 to 2016 in winter and spring. All of them

show strong longitudinal difference between North America (120°W-60°W) and Europe (10°W-26°E), with lower geopotential height, higher tropopause pressure (lower tropopause height), and greater stratospheric $O_3$ contribution over North America than Europe. The longitudinal gradients between North America and Europe are slightly weaker in spring than in winter. Skerlak et al. (2014) identified the deep STE hot spots along the western North America using the ERA-Interim reanalysis data set from the European Centre for Medium-Range Weather Forecasts (ECMWF) from 1979 to 2011.

Therefore, over North America, the stratospheric subsidence inside the polar vortex as well as deep stratospheric intrusion events results in a deeper and greater stratospheric $O_3$ influence on the tropospheric $O_3$ than that over Europe.

In terms of IAV, we refer to the Artic Oscillation (AO) – the primary mode of IAV in the troposphere during winter. Several studies have examined the mechanism for downward transport from the stratosphere to the troposphere and attributed changes in the strength of lower-stratospheric polar vortex to AO anomalies at the surface, with a positive AO phase linked

to a more isolated and stronger polar vortex (Ambaum and Hoskins, 2002; Perlwitz and Harnik, 2003) and lower tropopause heights. Lamarque and Hess (2004) found that the AO explains up to 50% of the IAV in tropospheric ozone over North America in January-March, but did not find any significant correlation in European sonde data, with similar results from the Model for OZone And Related chemical Tracers (MOZART) model. They argued that the correlation may be caused by the influence of the AO on its modulation of STE as well as transport of $O_3$ and its precursors. Kivi et al. (2007) found that

changes in the AO explained most of the tropospheric ozone trends in January–April, based on analysis of Arctic ozonesonde data. Figure 12 shows the longitudinal variations of simulated $O_3$ and AO correlation profiles averaged over 30°N and 80°N from 1000 hPa to 200 hPa. Over North America (120°W to 60°W), the correlation between simulated $O_3$ and the AO index stays low above 400 hPa, increases with increased pressure and reaches its maximum near surface around 90°W.  The correlation averaged over Europe (10°W-26°E) stays low above 400 hPa, increases slightly from 400 hPa to 700

hPa, then decreases sharply near the surface. This is similar to the correlations obtained from the ozonesonde profiles (Figure S3). The similarity of correlation patterns over sonde sites and their surrounding broader regions indicates that the AO-





related stratospheric subsidence is a large-scale phenomenon and also show a similar longitudinal variation between North America and Europe.

## 6: Conclusions and discussion

In this study we used $O_3$ and stratospheric $O_3$ tracer simulations from MERRA-2 GMI and observations from ozonesondes to investigate the interannual variations and vertical extents of stratospheric ozone influence on tropospheric ozone. Our work focuses on the winter and spring seasons over North America and Europe.

The model reproduces the observed interannual variations of tropospheric $O_3$ in the troposphere over North America except for the Pinatubo period from 1991 to 1995. The ozonesonde data show a negative ozone anomaly in 1992-1994 following the

Pinatubo eruption, with recovery thereafter. However, the simulated anomaly is about half the magnitude of the observed tropospheric ozone depletion. Over European regions, ozonesondes show a similar but weaker $O_3$ depletion, which was fully reproduced by the model. We use a stratospheric ozone tracer to gauge the impact of stratospheric ozone variations in different regions of the troposphere. Our results based on the stratospheric $O_3$ tracer suggest that the influence of the stratospheric IAV is significant in the middle to lower troposphere over North America, while over Europe, the stratospheric

influence is limited to the middle to upper troposphere. Over North America, the stratospheric subsidence insides the polar vortex as well as deep stratospheric intrusion events resulting in a deeper and greater stratospheric $O_3$ influence on the tropospheric $O_3$ than over Europe.

We examine the linkages of horizontal and vertical dynamical structures in the lower stratosphere to the contributions of stratospheric $O_3$ in the upper and middle troposphere. Our analysis suggests that the IAV of wave disturbances of the

westerlies likely affect the IAV of the regional distributions of prevailing wind patterns as well as the strength of STE flux. The IAV of stratospheric $O_3$ influence in the troposphere reflects a combined effect of the changes in the lower stratospheric $O_3$ concentration and in the 3-d dynamics, which may either cancel or reinforce each other, depending on their relative phases.

Our analysis provides an in-depth understanding of how dynamics influences the $O_3$ redistribution in the troposphere, and

reveals deficiencies in the transport produced by the input meteorological fields. The observed $O_3$ at 400 hPa over the North American sites show a similar $O_3$ depletion as that at 200 hPa, while in the model, the effect of lower stratospheric $O_3$ concentration seems masked by increased stratospheric-tropospheric flux, indicated by increased tropopause pressure accompanied by a stronger downward airmass flux in the model, especially between 50°N and 70°N. Therefore, the model underestimation of the observed $O_3$ depletion after the Mt. Pinatubo eruption over North America in the lower troposphere

could be due to the STE flux being too strong in the model for this region during that period. The assimilated MERRA-2 meteorological fields are significantly improved after the year 1998 when many higher-resolution meteorological observations are included in the assimilation (Bosilovich et al., 2015; Stauffer et al., 2019).



**Author contributions**

JL performed the data and model analysis and wrote the paper, JL and JMR conceived and planned the project and
participated in the numerous scientific discussions. LDO performed and provided the MERRA2-GMI simulation. ARD
provided insights on interpolation of model and data comparison. LDO and MAO helped on dynamical analysis of model
simulations and interpolation of the findings. LH prepared ozonesondes data. All authors provided critical feedback and
helped shape the research, analysis and manuscript.

**Acknowledgement**

All data used for this article can be obtained by contacting J. Liu (email:Junhua.liu@nasa.gov). I gratefully acknowledge the
financial support by NASA's Atmospheric Chemistry Modeling and Analysis Program (ACMAP) (grants NNX17AG58G).
Work was performed under contract with NASA at Goddard. Computer resources for the MERRA-2 GMI simulation were
provided by the NASA Center for Climate Simulation.

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

**Tables**

| Sonde station | (lat, lon) | Time | Freq (n/mon) |
|---|---|---|---|
| Alert | 82.50°N, 62.33°W | 1990-2017 | 4.1 |
| Eureka | 79.99°N, 85.94°W | 1993-2015 | 5.5 |
| Resolute | 74.72°N, 94.98°W | 1980-2017 | 3.1 |
| Churchill | 58.75°N, 94.07°W | 1980-2014 | 3.2 |
| Edmonton | 53.55°N, 114.10°W | 1980-2017 | 3.4 |
| GooseBay | 53.32°N, 60.30°W | 1980-2017 | 3.8 |
| Boulder | 40.00°N, 105.25°W | 1980-2017 | 3.0 |
| Wallops | 37.93°N, 75.47°W | 1985-2017 | 3.4 |

Table 1: The longitude, latitude, measurement time period and mean sampling frequency of the selected north American ozonesonde sites.

| Sonde station | (lat, lon) | Time | Freq (n/mon) |
|---|---|---|---|
| Ny-Aleasund | 78.93°N, 11.95°E | 1991-2013 | 7.1 |
| Sodankyla | 67.39°N, 26.65°E | 1989-2007 | 5.4 |
| Legionowo | 52.40°N, 20.97°E | 1980-2015 | 4.1 |
| Lindenberg | 52.21°N, 14.12°E | 1980-2014 | 5.0 |
| DeBilt | 52.10°N, 5.18°E | 1992-2014 | 4.3 |
| Uccle | 50.80°N, 4.35°E | 1980-2014 | 10.8 |
| Hohenpeissenberg | 47.80°N, 11°E | 1980-2017 | 10.0 |
| Payerne | 46.49°N, 6.57°E | 1980-2014 | 11.2 |
| Madrid | 40.47°N, 3.58°W | 1995-2015 | 3.6 |

Table 2: The longitude, latitude, measurement time period and mean sampling frequency of the selected European ozonesonde sites.

| | North American stations | European stations |
|---|---|---|



|  | (1990-2016) | | (1990-2016) | | (1992-2015) | |
|---|---|---|---|---|---|---|
|  | DJF | MAM | DJF | MAM | DJF | MAM |
| r (200 hPa - 400 hPa) | 0.52 (0.52) | 0.64 (0.68) | 0.31 (0.71) | 0.13 (0.61) | 0.67(0.79) | 0.23 (0.64) |
| r (200 hPa - 700 hPa) | 0.25 (0.05) | 0.46 (0.41) | 0.32 (0.12) | 0.27 (0.17) | 0.43(0.35) | 0.39(0.29) |

**Table 3: correlation coefficients of ozone between 200 hPa and 400 hPa, 200 hPa and 700 hPa in observations and simulations. The numbers in parentheses are correlation coefficients for simulations.**

**Figures**

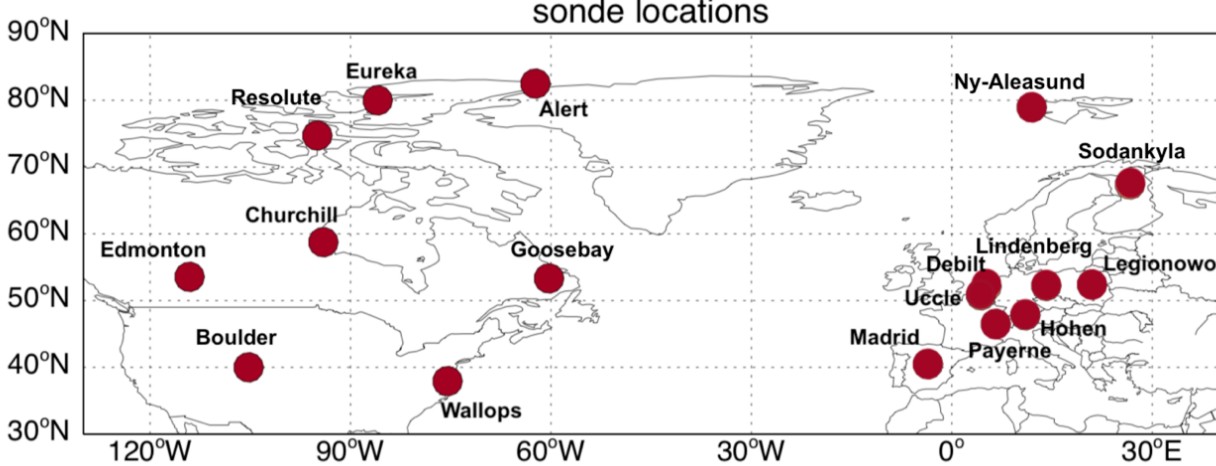

**Figure 1: Map of ozonesonde sites selected in this study.**



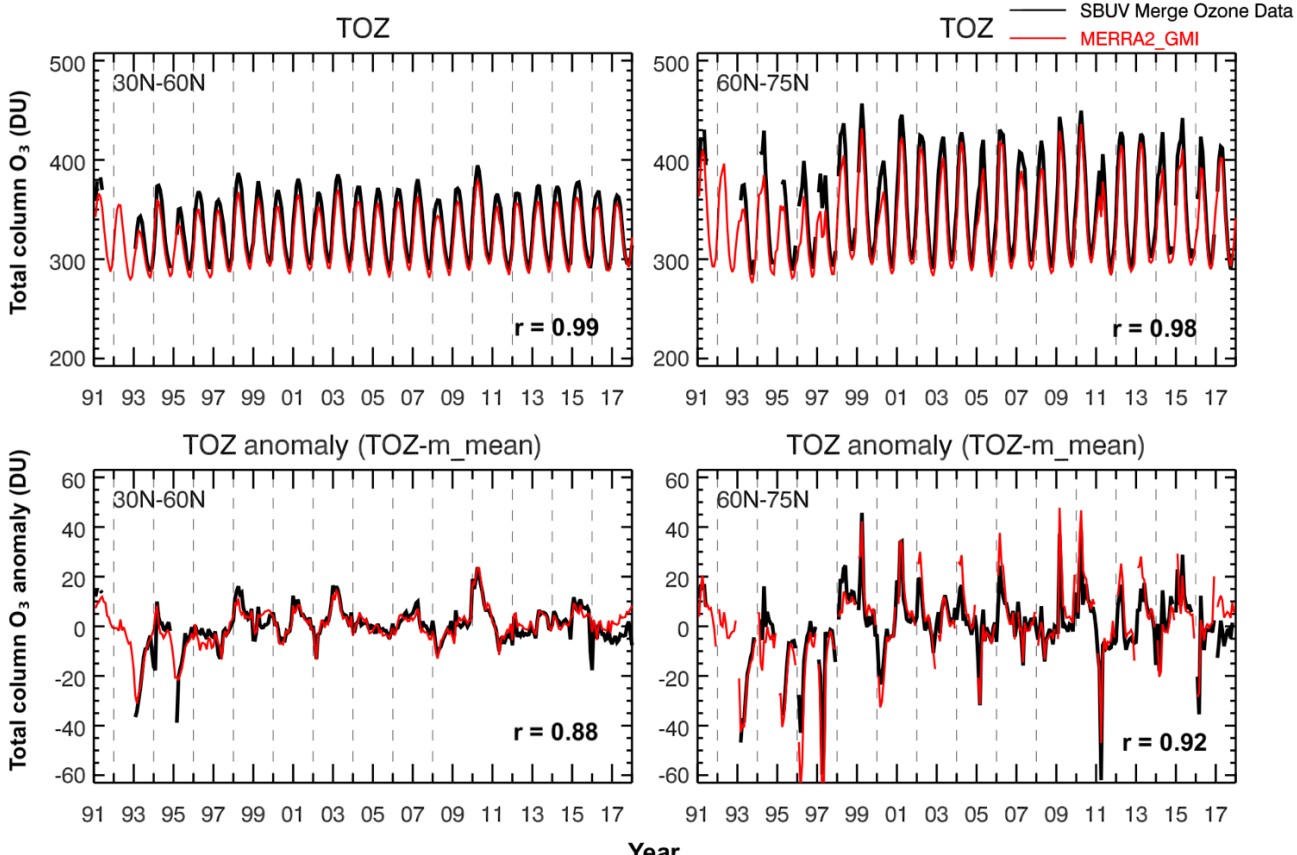

**Figure 2: Monthly zonal mean of total column ozone (top) and its anomalies (bottom) averaged over (left) 30°N-60°N and (right) 60°N-75°N from the observations of the SBUV version 8.6 merged total ozone datasets (black lines) and the MERRA2-GMI simulations (red lines) from 1991 to 2016. The anomalies are calculated by removing the monthly mean averaged from 1991 to 2016.**

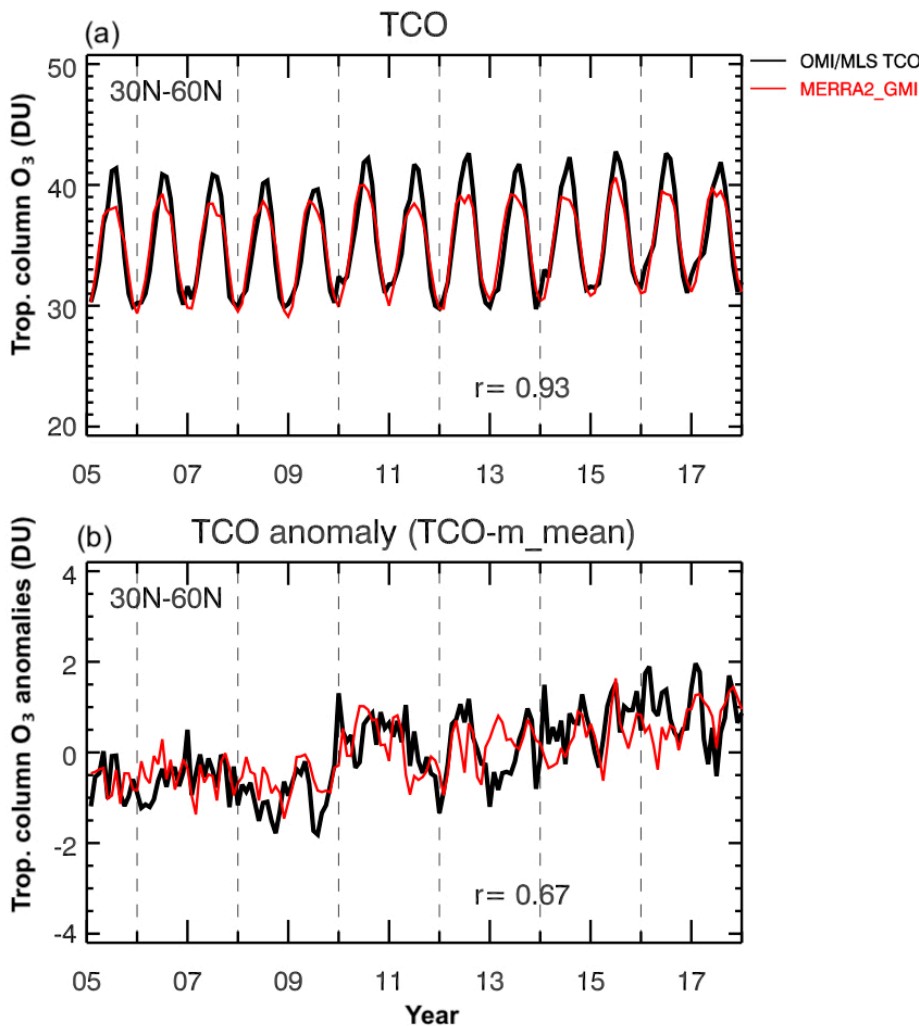


**Figure 3: Monthly zonal mean of total column ozone (top) and its anomalies (bottom) averaged over 30°N-60°N from the observations derived from OMI/MLS residual analysis (black lines) and the MERRA2-GMI simulations (red lines) from 2005 to 2017. The anomalies are calculated by removing respective monthly mean averaged from 1991 to 2017.**





**Figure 4: Monthly mean of observed (black) and simulated (red) ozone anomalies averaged over selected 17 ozonesonde stations at 200 hPa, 400 hPa and 700 hPa. The anomalies are calculated by removing the respective monthly mean from 1990 to 2016. The Shaded area represents the 95% confidence interval (CI) of observed mean, which is calculated by multiplying the standard error of observations by 1.96. The O₃ anomalies shown are smoothed using a 6-month running mean.**



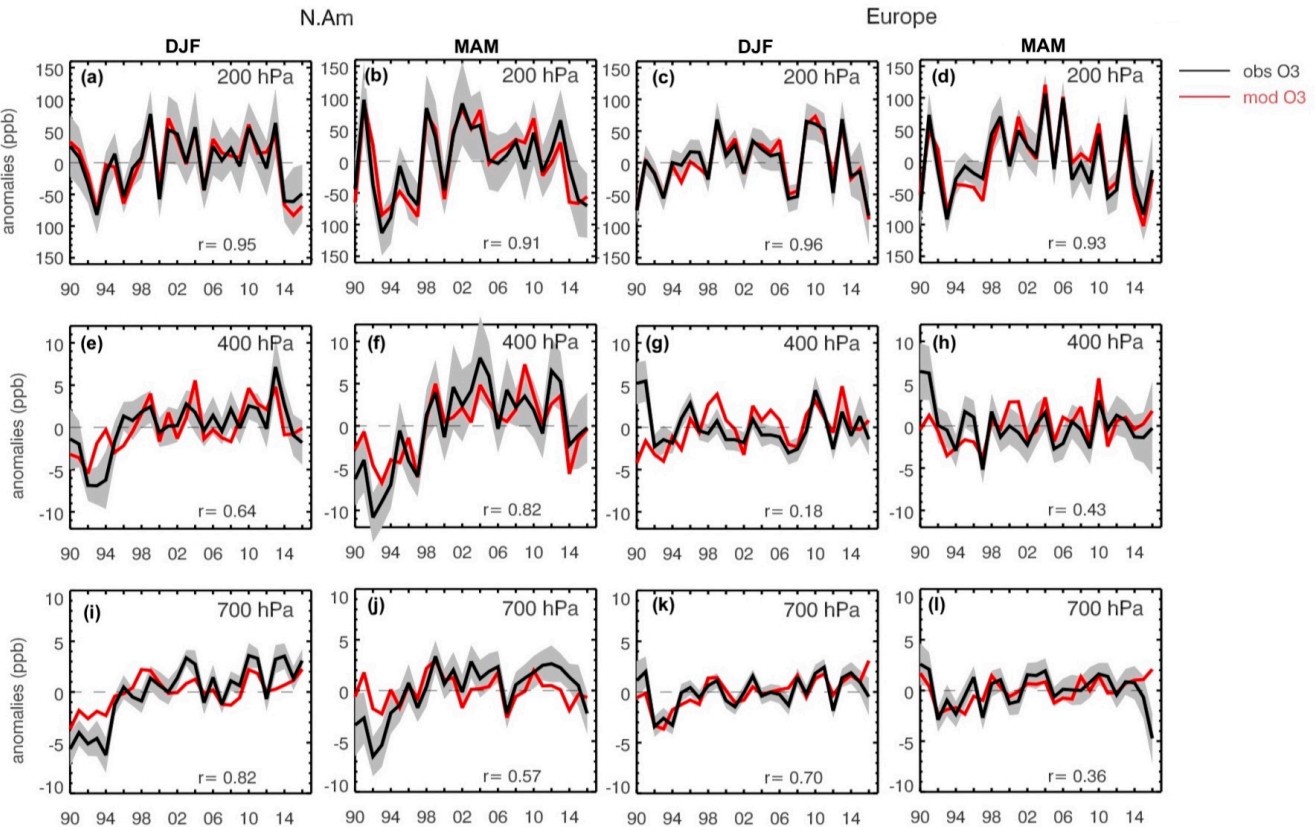

**Figure 5: Time series plots of observed (black) and simulated (red) ozone anomalies (unit: ppb) at 200 hPa (top), 400hPa (middle) and 700 hPa (bottom) averaged from selected ozonesonde sites over North America and Europe in winter and spring seasons from 1990 to 2016. The anomalies are calculated by removing the seasonal mean averaged from 1990 to 2016. The shaded area represents the 95% confidence interval (CI) of observed mean, which is calculated by multiplying the standard error of observations by 1.96.**





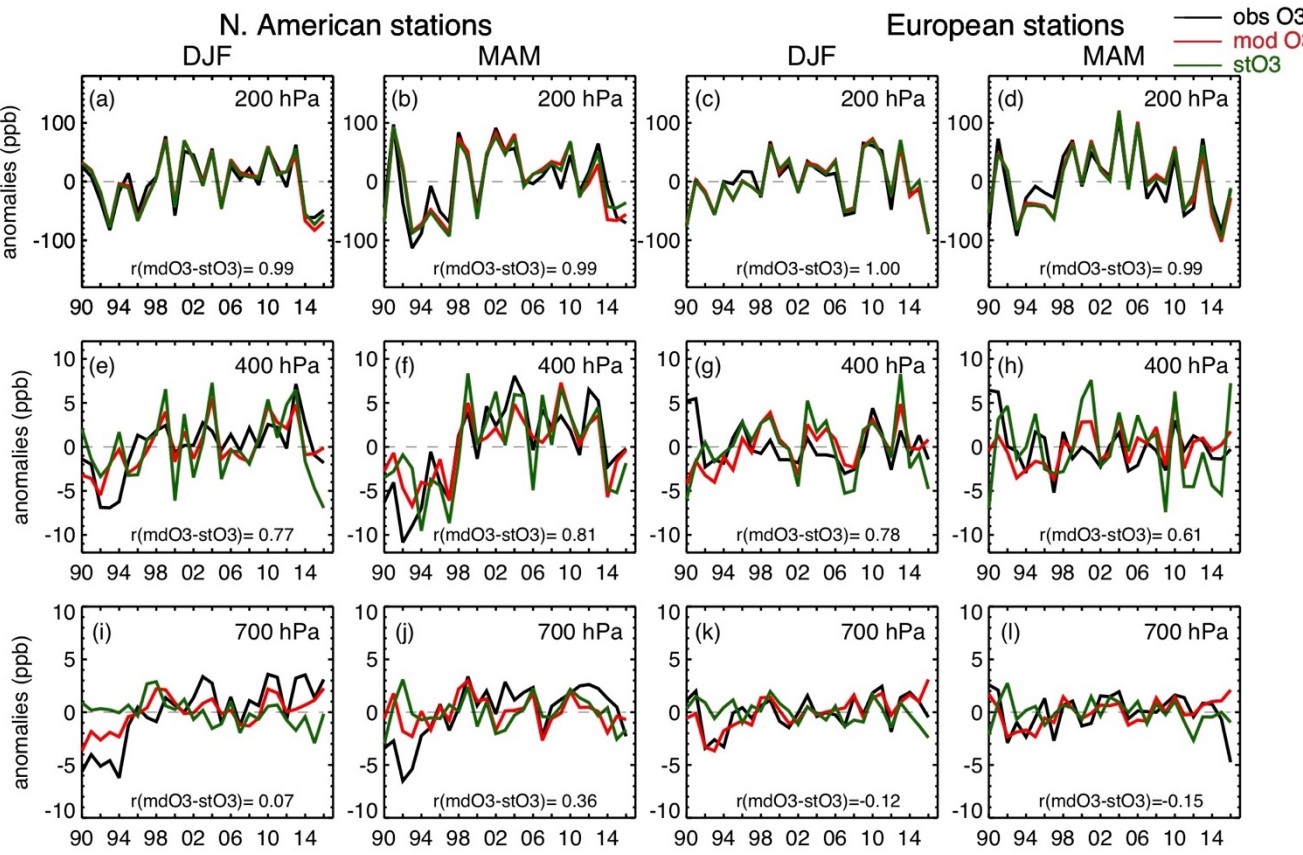


**Figure 6: Similar to Figure 5, but adding the simulated StratO₃ anomalies (green). The correlation coefficients between simulated O₃ and StratO₃ are shown in text.**

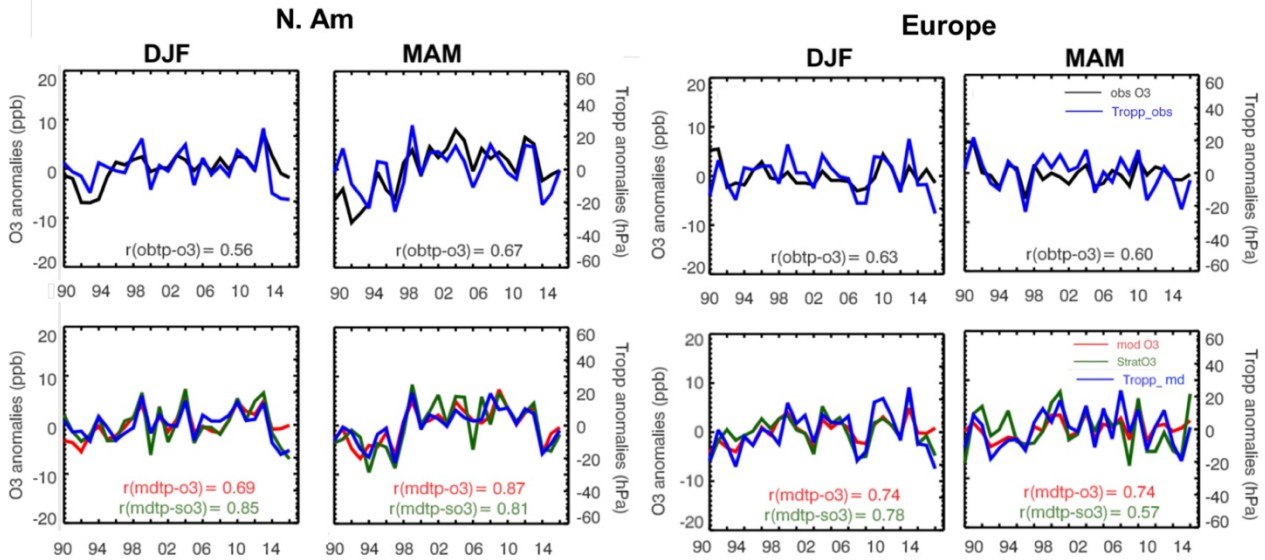

**Figure 7: (top) Time series of the observed O₃ mixing ratio anomalies at 400 hPa and the tropopause pressures derived from observed O₃ profiles averaged over the North American and European sites in winter and spring. Their correlation coefficients are shown in black text. (bottom) Time series of the simulated O₃ and StratO₃ anomalies at 400 hPa with the tropopause pressures derived from simulated O₃ profiles, with the respective correlation coefficients shown in red and green text.**


**Figure 8: : Spatial maps of simulated StratO$_3$/O$_3$ ratio (1st row) and its anomaly (2nd row) at 400 hPa, Airmass flux (3rd row) and its anomaly (4th row) at 250 hPa in winter (left) and at 300 hPa in spring (right) of 1993. 250 hPa and 300 hPa are the closest model pressure levels to the area averaged tropopause pressure between 30°N and 80°N. Black thin arrows in 1st row represents the prevailing wind pattern at 400 hPa. Red thick lines indicate the approximated jet locations, where the strongest winds are.**






**Figure 9: Similar to Figure 8 but for year 1998.**





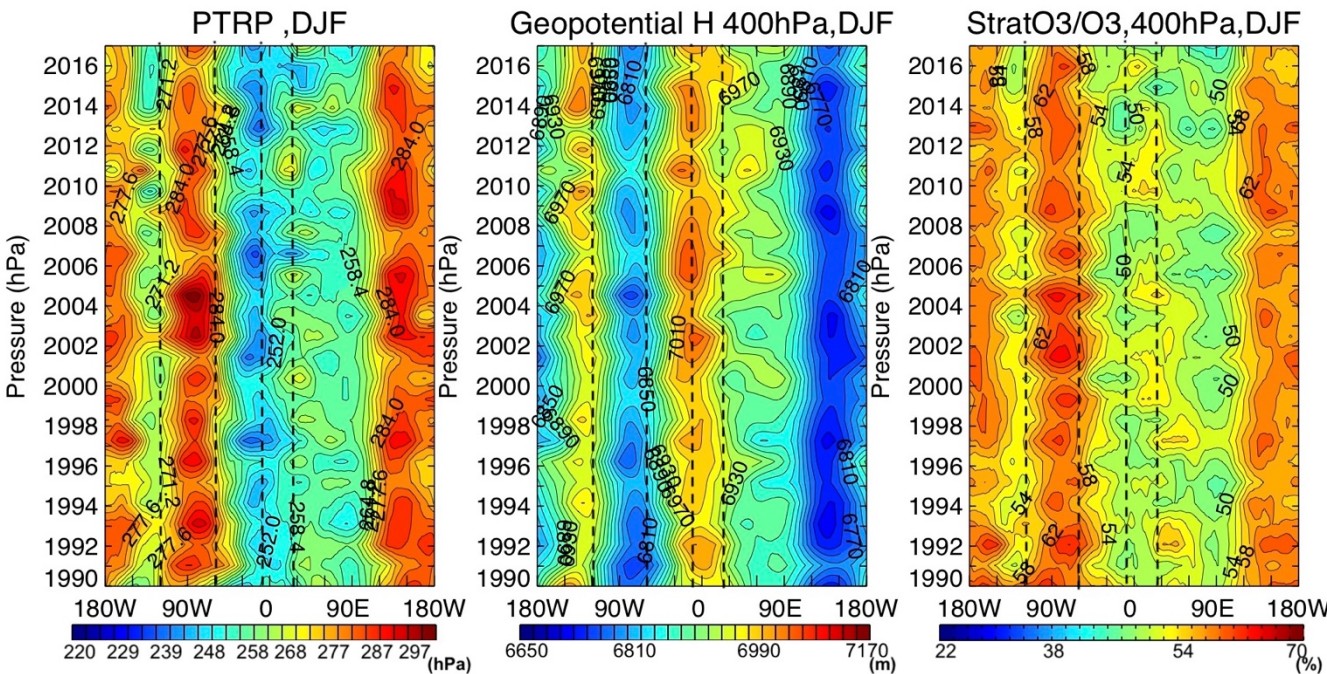


**Figure 10: Latitudinal average between 30°N to 80°N of (left) the tropopause pressure; (middle) the geopotential height at 400 hPa; (right) the StratO₃/O₃ ratio at 400 hPa along each longitude from 180°W to 180°E from 1990 to 2016 in winter (DJF). Dashed lines indicate the longitudinal range for the North American region (120°W-60°W) and the European region (10°W-26°E).**





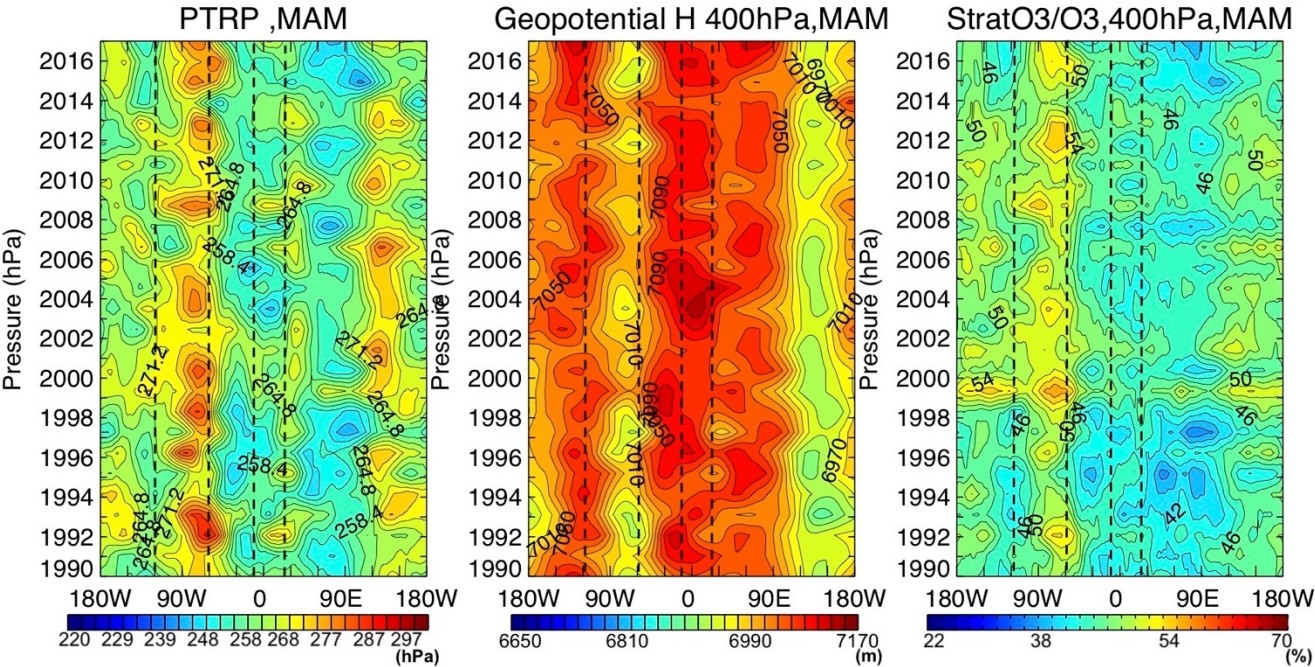

**Figure 11: Latitudinal average between 30°N to 80°N of (left) the tropopause pressure; (middle) the geopotential height at 400 hPa; (right) the StratO₃/O₃ ratio at 400 hPa along each longitude from 180°W to 180°E from 1990 to 2016 in spring (MAM). Dashed lines indicate the longitudinal range for the North American region (120°W-60°W) and the European region (10°W-26°E).**

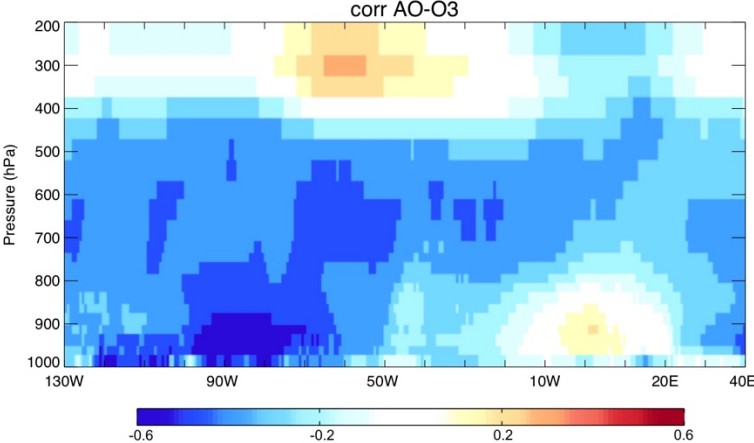

**Figure 12: Longitudinal variations of simulated O₃ and AO correlation profiles (r) averaged over 30°N and 80°N from 1000 hPa to 200 hPa.**