# Peer review of "Stratospheric impact on the Northern Hemisphere winter and spring ozone interannual variability in the troposphere"

_Atmospheric Chemistry and Physics, 2019_

## Referee Comment (RC1) · Anonymous Referee #1 · 16 Nov 2019

Overview: This paper uses modeled and observed ozone to examine the interannual variation of the impact of stratospheric ozone on tropospheric concentrations and is restricted to mid to high latitudes in the NH during winter and spring. The authors conclude that the model well reproduces the interannual variations in tropospheric ozone, except over North America following the eruption of Mt. Pinatubo. They infer that the STE was too strong over NA after the Pinatubo eruption. The paper will be suitable for publication, but I recommend revision prior to acceptance, after the authors have considered the questions noted below.

Question 1: The authors state that the stronger and deeper stratospheric contributions

[Figure]

in the tropospheric O3 variability shown by the model is related to the ozonesondes being closer to the polar vortex in winter over NA than over Europe. This doesn't make sense to me. Does it mean that you're effectively comparing apples and oranges, in that you're looking at different meteorological regimes when looking at your NA data vs your European data? The text makes it sound like the ozonesondes are somehow controlling what the model does.

Question 2: The Orbe 2017 paper referenced talks about multiple version of a replay simulation, and discusses various deficiencies in the large-scale transport depending on how the simulation was done. Which one of the runs discussed in the Orbe paper is this study using? Or, because it seems this is a higher horizontal resolution run than discussed in Orbe et al, 2017, is it something completely different? My concern is that the Orbe paper talks about potential issues (i.e., regarding age of air in particular) regarding the replay simulations, so have you picked a version of the model that would best represent overall transport?

Question 3, discussion of figure 4 tropospheric comparison. The authors states that the phase is in agreement but the magnitude is underestimated by the model for the observed anomalies. (and, do you calculate the anomalies from the individual stations and then average, or from the averaged ensemble of 17 stations? This should be stated before the figure is presented.) I think really you mean sign is in agreement rather than phase. I also don't see that in general that the absolute value is underestimated by the model. At 700 mb, the model and obs don't agree on the sign for the period from 2012-2015. At 400 mb, they don't agree on the sign for 1990-end of 1991. At 400 mb, there is an underestimate sometimes, and an overestimate from 1997-2001. I also don't understand the statement that both obs and simulations show the largest interannual variations in winter and spring. Am I supposed to be able to discern that from Figure 4? Perhaps that statement shouldn't be made until you've presented figure 5. And, in the caption of figure 4, please say what the red and black numbers are supposed to mean.

[Figure]

Question 4, discussion of figure 5. The authors state that, for 200 mb, the IAV is larger over NA than Europe, and larger in spring than winter. These appear to be qualitative statements. Do you have a way to calculate a value for IAV (i.e., perhaps the standard deviation of your anomalies)? It would then be possible to apply some sort of statistical test to assess whether there really is a regional or seasonal difference.

Question 5, The author's state that the correlation between polar winter 150 mb temps and 200 mb ozone anomalies being lower in spring is "consistent with our understanding of the impact of temperature variations on the formation of polar stratospheric clouds and polar vortex isolation with reduced transport of o3 from the tropics at low temperatures....". I personally don't follow this at all. Are you trying to explain why there is a correlation, or why the correlation is different between winter and spring?

Question 6, I think you need a quantifiable definition of what you mean by IAV in order to compare where it is larger or smaller in different seasons or in different regions. The paper is written as though IAV is the same as the deviation (anomaly) from the seasonal mean. One then has to determine the interannual variations from looking at wiggles in anomaly plots.

Question 7: Discussion of Table 3, Please explain how, from looking at the correlation coefficients in Table 3, that one concludes that 27% of the NA interannual variation is related to 200 mb changes in winter.

Question 8: Discussion of Figure 6, Mt Pinatubo erupted in June 1991. Your 700 mb DJF NA plot shows a large difference between the red, black and green lines for 1990. What are you defining as the "Pinatubo period" and do you keep 1990 in your re-calculations of strato3-o3 correlation when you say you omit the Pinatubo period?

Question 9: around line 260-265 it states that anomalies in strato3 diverge from simulated o3 near the end of the period, and looking at figure 5, that seems to be around 2012. Do precursors really become significantly important only in the past decade?

[Figure]

Question 10: If you separate the analysis more finely than simply Europe vs NA, and compared comparable latitudes, do you come to the same conclusions? How different are Madrid and Wallops? Your NA comparison includes more high latitude stations than your European one does. Is it longitude you're finding differences between, or latitude?

Question 11: On line 306, replace "changes" with "relationship" Your plot shows snapshots of winter and spring 1993, not differences (or changes).

Question 12: Final paragraph, the implication here is that the underlying meteorology was deficient over NA in the early period, but perhaps not over Europe. What would be the reason for that? And, can you look at any other fields in the model/sonde comparisons to assess whether this is the issue (maybe tropopause pressure, or the temperature from the radiosonde that flew with the ozone sonde?

specific comment: please change "amplitude" on line 194 to "magnitude".

---

## Referee Comment (RC2) · Anonymous Referee #2 · 26 Nov 2019

Liu et al. use model simulations of ozone and a stratospheric ozone tracer together with observations from ozonesondes to investigate the interannual variation of ozone and the vertical extent of the impact of stratospheric ozone on tropospheric ozone. Before the simulations are used for the analyses their quality is checked by first comparing the simulations to measurements.

I am confident that the study itself is important and deserves to be published, however, I am not happy with how the result from these study are presented. The manuscript in its present form is confusing and needs thorough structuring and a clear line. From the current manuscript is not clear what the major focus of this study is: Do you want

to evaluate the model or do you want to investigate the stratospheric impact on the NH winter and spring interannual variability in the troposphere as it is stated in the title. The manuscript in its current form has a stronger focus on the evaluation of the model than on the analyses of the interannual variability.

Further, a lot of information is packed into the figures and thus makes it quite hard to follow and get the major results through. I would suggest major revisions before the manuscript can be published.

Specific comments: P1, general: Why is it important to look at the interannual variation? What are the unanswered questions? The motivation for this study is not clear. In the introduction (P2, 58-59) a motivation is given. Something like this could be repeated in the abstract.

P1, L1: How long is the model run? That should be mentioned here.

P1, L29-30: Why should ozone sondes be closer to the polar vortex? This sentence is somewhat weird and misleading and thus should be rephrased.

P2, L44: What exactly are these "replay" simulations? This should be explained. What atmospheric conditions or initial conditions have been assumed for this simulation?

P2, L48: Which parameters exactly? Can you give some examples?

P3, L75ff: Here you give a better description of the aim of this study. Something like this should be also added in the abstract, so that it also there becomes more clear why it is important to investigate these processes.

P3, Section: A comparison for each station would also be quite useful to understand local differences and which stations/locations maybe mess up the mean.

P5, L135ff: The comparison to the satellite data has not been mentioned in the abstract or introduction. Why? If it is a part of this study it should be mentioned there. Why do you this comparison in the first place? Is this really necessary? You anyway compare

the model simulations to sonde data so. Therefore, I do not understand what additional information is gained by doing an additional comparison. Especially, if your focus is not on the evaluation of the model but on the investigation of the impact of stratospheric ozone on tropospheric ozone.

P5, L154ff: Reference to the figure is missing.

P7, L205ff: I cannot follow how you derive this conclusion. Which season and time periods are you referring to? How have the numbers in percent been derived?

P8, L228: What exactly is the StratO3 tracer? What is included in the diagnostic? How is it calculated? Is this simply the stratospheric O3 flux?

P8, L234: Where exactly do we see this in Figure 6?

P9, L266ff: Also here it is not clear how the numbers in percent have been derived.

P9, L267-267: Here an important result is given, but it gets somehow lost in the discussion of the differences between the model simulations and observations.

P9, L269: Reference? Has this relations seen before? Has this relation already been discussed somewhere else?

P10, L298-299: This sentence is too complicated and should be rephrased. Maybe it would be better to split this sentence also into two sentences.

P10, L308: It would be worth to more clearly state that because of the different tropopause heights different pressure levels are shown in the figures.

P10, L315: How it the air mass flux derived/calculated?

P10, L320: not shown? Or is this shown? Can this be seen when comparing 1993 to 1998?

P10, general: In the introductory part of this section StratO3/O3 distinction based on PV is mentioned, but in the analyses the air mass flux is used.

P11, L327: Here four panels are given, but only 2 panels show the 400 hPa level.

P11, L330: Why is there less dynamic perturbation?

P12, L363: Why are these three parameters used? What is the connection between these? This is not really discussed. Wouldn't it then be easier to just show StratO3/O3?

P12, L383: maximum? Shouldn't it read minimum? Generally, I have the feeling that in this paragraph the description does not agree with the figure shown.

P13, L396: This is not clear. How does the Pinatubo eruption deplete ozone? Do you mean in the troposphere or the stratosphere and by which process?

P13, L410-411: This does not become comprehensible from what is shown in the manuscript.

Figure 2 and 3: Are these figures really useful? Especially, since later anyway the simulations are compared to ozone sonde data. This part of the study could (if required) be provided in the supplement.

Figure 4: What does the reader gain from this Figure? Is there any more information gained when comparing observations from all stations with the model simulation?

Figure 5, 6, and 7: I would suggest to split these by North America and Europe and discuss the regions separately. As you do it now, you compare different pressure levels, seasons and regions and it gets really hard to follow since you also above all that additionally discuss the differences between model simulation and observations.

Figure 8, 9: Again too many panels and too many things discussed at the same time. I would suggest to solely show the anomalies in the figure and to provide the airmass flux in the supplement.

Technical comments: P2, L18: add "of O3" after input and maybe use a different wording for "input", e.g. entrainment.

P2, L47: "in so doing" → "in doing so"?

P4, L99: present = 2019? It would be better to clearly state the year here.

P4, Section 4 header: remove colon.

P4, Section 4.1 header: remove full stop after title.

P7, L219: space between "correlation" and reference of "Terao" missing.

P7, Section 4.3 header: Remove colon.

P12, L360: "impact on tropospheric O3 from the upper to lower troposphere" → not clear. Please rephrase the sentence.

P13, Section 6 header: remove colon.

Figure 8 and 9: Panel labelling with a,b,c. . ... is missing.

Figure 8: Adjust both columns so that they are next to each other at the same height. At the moment there is a shift between the columns.

Figure 10 and 11: 180 W on the right side of the x-axes should read 180 E.

Figure 10 and 11: To use white dashed lines instead of black dashed lines would increase the readability.

Figure 12: Also here North America and Europe should be marked.

---

## Referee Comment (RC3) · Anonymous Referee #3 · 18 Dec 2019

The paper compares the 1990-2015 ozonesonde observations at 8 North American and 9 European sites with CCM output of tropospheric ozone levels to study the stratospheric impact on the observed tropospheric ozone concentration time series. The (total + tropospheric) ozone output of the model is first validated by comparison with satellite ozone retrievals. Making use of a model stratospheric ozone tracer, the impact of STE on tropospheric ozone is assessed, together with the analysis of model wind patterns and airmass fluxes.

GENERAL COMMENTS

The study is scientifically sound and takes into account all relevant literature. The

analysis is detailed and all relevant aspects are considered. The presentation is clear, although somewhat verbose at some locations, and follows a very logical structure. It therefore deserves publication in ACP, if some remaining issues can be described better or clarified. These are summed up here below.

SPECIFIC COMMENTS

* From the text (page 4, lines 114-120), it is not clear how the stratospheric ozone tracer (StratO3) is defined. Please be more specific on this important variable of your analysis.

* On page 5, lines 147-150: please, be more quantitative when comparing the magnitude, IAV and trend of the tropospheric ozone satellite retrieval and model replay simulation. More in general, I agree with reviewer 1 that, throughout the entire manuscript, you should quantify the comparison of "IAV" between two datasets.

*On Page 5, lines 154-156, please describe more clearly how the ozone anomalies are calculated. For instance, for every ozonesonde site, you first calculate the monthly anomalies, and then you calculate the monthly mean of those monthly anomalies for all sites together? What does the 95% confidence interval represents ? The site to site variability with or without the variability within one month at a given site?

*Coming back to the previous point: quantify the statements on page 6, lines 168-169: "Both observations and simulations show the largest interannual variations in the winter and spring, when the strongest IAVs occur" and on page 6, lines 176-177: "The IAV of ozone is larger over North America then over Europe, and larger in spring than in winter".

* In sect 4.1, in which you describe Fig. 4, it should be mentioned that the comparison between ozonesonde data and model simulation decrease with increasing pressure and why this is the case.

* Page 6, lines 184-188: I do not understand the link between the winter polar 150 hPa

AVERAGED temperature – 200 hPa O3 IAV correlation and PSC formation, which only happens at very low stratospheric temperatures (< -80°C).

* Page 7, lines 206-209: where do these explained variances come from (in Table 3, only correlations are shown)? Please explain. Same comment for the percentages for the explained variations, mentioned on Page 8, line 234, and page 9, lines 265-267.

* Page 9: why are you using the alternative definition of tropopause pressure by Browell et al. (1996)? Is this tropopause identical to the ozonopause? What is the effect of this choice for the tropopause (compared to the thermal tropopause, as defined by the WMO) on the mentioned correlations with the IAV of O3 and stratO3?

* Page 12, lines 372-388: the analysis of the correlations between AO and ozone is not very convincing. First of all, please mention the months for which Fig. 12 is constructed (DJF and/or MAM?). Secondly, on which ground do you classify the correlation profiles (with low correlation coefficients after all) in Fig. 12 as significantly different between North America and Europe? And similar between sonde and model data in Figure S2?

TECHNICAL CORRECTIONS

* Pag 1, line 29: remove the ' after ozonesondes

* Page 2, line 46-47: replace "In so doing" with "In doing so".

* Page 5, before Section 4: Here, you can add that some features in tropospheric ozone are well reproduced (e.g. 2015), while others not (e.g. 2013) and that those differences will be analyzed further in the paper.

* Page 9, after line 269: please mention here that the longitudinal difference in dynamics between North America and Europe will be further analyzed in Sect. 5.2.

* Page 10, line 310: replace "asterisks" by "lines" (referring to Fig. 8).

* Page 13, line401: replace "resulting" with "result".

* Please remove the : in the section titles (e.g. 6: Conclusions and discussion)

* Please acknowledge the data repositories properly for the ozone data used (ozonesondes: WOUDC, SBUV, OMI, etc.).

---

## Short Comment (SC1) · 23 Dec 2019

This is an interesting new article on the role of the of the stratosphere on tropospheric ozone interannual variability during Northern Hemisphere winter and spring (when the STE flux is at a maximum). We however feel that our most recent study that looks at the stratospheric influence on tropospheric ozone should additionally be cited within the introduction:

"Characterising the seasonal and geographical variability in tropospheric ozone, stratospheric influence and recent changes" by Ryan S. Williams et al. (2019) (https://www.atmos-chem-phys.net/19/3589/2019/)

We would suggest adding a citation to this paper either on P2, L38:

"Stratosphere-troposphere exchange (STE) has been shown to impact the tropospheric ozone distribution (e.g., Terao et al., 2008; Hess et al., 2015; Holton et al., 1995)."

Or alternatively on P2, L50:

"STE has been widely studied for several decades (Danielsen, 1968; Holton et al., 1995; Olsen et al., 2002; 2003; 2013; Stohl et al., 2003a; 2003b; Sprenger and Wernli, 2003; Thompson et al., 2007; Lefohn et al., 2011; Skerlak et al., 2014)".

Since our study does not look at STE explicitly (only implicitly using tagged stratospheric ozone tracers from the EMAC and CMAM CCMs), a citation on L38 would be more applicable in our view.

Furthermore, we feel that a mention to nudged, specified-dynamics CCM simulations should be later included in the introduction, in addition to free-running CCM simulations and CTMs (P2-3, L61-72), as a useful tool for assessment of the stratospheric influence on tropospheric ozone (using stratospheric tagged ozone tracers). Compared with free-running CCMs, "the influence on composition of dynamical biases and differences in variability between the reanalysis and the models can be assessed" - Morgenstern et al. (2017), P648 (https://www.geosci-model-dev.net/10/639/2017/). This point could also be made in highlighting the role of constraining the dynamics on influencing the distribution of model composition fields.

---

## Author Comment (AC1) · 15 Feb 2020

**Response to reviews on "Stratospheric impact on the Northern Hemisphere winter and spring ozone interannual variability in the troposphere"**

**by Junhua Liu et al.**

We thank the three reviewers for their helpful comments and Ryan Williams for his interactive comment. We have addressed all comments in detail below and have clarified the text in the relevant sections.

In the following, we address the concerns raised by all the reviewers. Reviewers' comments are italicized.
* * *
*Anonymous Referee #1*

*Overview: This paper uses modeled and observed ozone to examine the interannual variation of the impact of stratospheric ozone on tropospheric concentrations and is restricted to mid to high latitudes in the NH during winter and spring. The authors conclude that the model well reproduces the interannual variations in tropospheric ozone, except over North America following the eruption of Mt. Pinatubo. They infer that the STE was too strong over NA after the Pinatubo eruption. The paper will be suitable*

*for publication, but I recommend revision prior to acceptance, after the authors have considered the questions noted below.*

*Question 1: The authors state that the stronger and deeper stratospheric contributions in the tropospheric $O_3$ variability shown by the model is related to the ozonesondes being closer to the polar vortex in winter over NA than over Europe. This doesn't make sense to me. Does it mean that you're effectively comparing apples and oranges, in that you're looking at different meteorological regimes when looking at your NA data vs your European data? The text makes it sound like the ozonesondes are somehow controlling what the model does.*

Thanks a lot for the comments by the first reviewer. The text has been modified to avoid the confusion. Figure 9 and 10 in revised manuscript show that there are strong longitudinal variations in NH (averaged between 30°N to 80°N) meteorology (tropopause pressure, geopotential heights), which results in the longitudinal variations of stratospheric $O_3$ contribution between N. America and Europe. Please see below for the modified text:

Our analysis of the MERRA2 assimilated fields shows strong longitudinal variations in meteorology over northern hemisphere (NH) mid-high latitudes, with lower tropopause height and lower geopotential height over North America than Europe. These variations associated with the relevant variations in the location of tropospheric jet flows are responsible for the longitudinal change in the stratospheric $O_3$ influence and result in a deeper and greater stratospheric $O_3$ influence on the tropospheric $O_3$ over North America than that over Europe.

*Question 2: The Orbe 2017 paper referenced talks about multiple version of a replay simulation, and discusses various deficiencies in the large-scale transport depending on how the simulation was done. Which one of the runs discussed in the Orbe paper is this study using? Or, because it seems this is a higher horizontal resolution run than discussed in Orbe et al, 2017, is it something completely different? My concern is that the Orbe paper talks about potential issues (i.e., regarding age of air in particular) regarding the replay simulations, so have you picked a version of the model that would best represent overall transport?*

We are referring the Orbe et al (2017) paper to explain the detailed description of the "replay" methodology. The runs discussed in the Orbe paper are performed at a coarser resolution. Neither of them is the one used in our study. The simulation used in our study has the similar setting as RAs3, which best represents overall transport. The text has been modified as below:

We use a replay simulation ([http://acd-ext.gsfc.nasa.gov/Projects/GEOSCCM/MERRA2GMI)](http://acd-ext.gsfc.nasa.gov/Projects/GEOSCCM/MERRA2GMI)) of the GEOSCCM with the Global Modeling Initiative (GMI) chemical mechanism (Strahan et al., 2007;Duncan et al., 2007) for trace gas chemistry, which includes a complete treatment of stratospheric and tropospheric chemistry, and the Goddard Chemistry Aerosol Radiation and Transport (GOCART) module (Chin et al., 2002;Colarco et al., 2010) for aerosols. The replay simulation follows the replay methodology as described in Orbe et al. (2017) and uses the RAs3 setting, which best represents overall transport. The model reads in the three-hourly time-averaged output of MERRA-2 meteorology (U, V, T, pressure) and recomputes the analysis increments, which are used as a forcing to the meteorology at every time step over the 3 h replay interval. More detailed information on replay methodology can be found in Orbe et al. (2017). The replay simulation is run at a MERRA-2 native resolution of ~50 km in the horizontal dimension and 72 vertical levels. This replay simulation is referred to as the 'MERRA2-GMI" simulation.

*Question 3, discussion of figure 4 tropospheric comparison. The authors states that the phase is in agreement but the magnitude is underestimated by the model for the observed anomalies. (and, do you calculate the anomalies from the individual stations and then average, or from the averaged ensemble of 17 stations? This should be stated before the figure is presented.) I think really you mean sign is in agreement rather than phase. I also don't see that in general that the absolute value is underestimated by the model. At 700 mb, the model and obs don't agree on the sign for the period from 2012- 2015. At 400 mb, they don't agree on the sign for 1990-end of 1991. At 400 mb, there is an underestimate sometimes, and an overestimate from 1997-2001. I also don't understand the statement that both obs and simulations show the largest interannual variations in winter and spring. Am I supposed to be able to discern that from Figure 4? Perhaps that statement shouldn't be made until you've presented figure 5. And, in the caption of figure 4, please say what the red and black numbers are supposed to mean.*

We agree with the reviewer 2 that Figure 4 did not provide more useful information by comparing observations from all stations with the model simulation. We therefore removed Figure 4 and section 4.1.

*Question 4, discussion of figure 5. The authors state that, for 200 mb, the IAV is larger over NA than Europe, and larger in spring than winter. These appear to be qualitative statements. Do you have a way to calculate a value for IAV (i.e., perhaps the standard deviation of your anomalies)?*

*It would then be possible to apply some sort of statistical test to assess whether there really is a regional or seasonal difference.*

Thanks a lot for the reviewer's suggestion on statistical analysis. We calculated the standard deviations of the anomalies to support our arguments of IAV. We also performed several statistical F-test to assess the equality of variance (standard deviation) for the selected anomalies. The significance of F-test is a value in the interval [0.0, 1.0]; a small value (< 0.2) indicates that the selected two datasets have significantly different variances. Below are two tables to assess whether there is a significant difference in the IAVs 1) between North America and Europe, 2) between DJF and MAM. The objective of our paper is quantifying the stratospheric $O_3$ influence on the tropospheric $O_3$ IAV, the seasonal or regional difference of $O_3$ IAV is not the focus of our paper. We therefore add those tables into supplementary materials. Corresponding discussions are added into text.

| | | DJF | MAM |
|---|---|---|---|
| *200 hPa* | $Std_{na}$ ($Std_{eu}$) | 44 (44) | 57 (54) |
| | F-test | 0.99 | 0.82 |
| *400 hPa* | **$Std_{na}$ ($Std_{eu}$)** | **3.08 (2.34)** | **4.94 (2.54)** |
| | F-test | **0.17** | **0.001** |
| *700 hPa* | **$Std_{na}$ ($Std_{eu}$)** | **2.94 (1.59)** | **2.56 (1.73)** |
| | F-test | **0.002** | **0.05** |

Table R1: Standard deviations and F-test statistics of the observed $O_3$ anomalies over N. American sites ($Std_{na}$) and European sites ($Std_{eu}$), to assess whether there is significant regional difference in the amplitude of IAVs between North American and European sites.

At 200 hPa, there is not significant regional difference in the magnitude of $O_3$ IAV between North America and Europe in both seasons. At 400 hPa and 700 hPa, ozonesonde observations show significantly greater IAV over North America than Europe in both seasons.

| | | North America | Europe |
|---|---|---|---|
| *200 hPa* | $Std_{djf}$ ($Std_{mam}$) | **44 (57)** | **44 (54)** |
| | F-test | **0.19** | **0.28** |
| *400 hPa* | $Std_{djf}$ ($Std_{mam}$) | **3.08 (4.94)** | 2.34 (2.54) |
| | F-test | **0.02** | 0.69 |
| *700 hPa* | $Std_{djf}$ ($Std_{mam}$) | 2.94 (2.56) | 1.59 (1.73) |
| | F-test | 0.5 | 0.66 |

Table R2: Standard deviations and F-test statistics of the $O_3$ anomalies in DJF ($Std_{djf}$) and MAM ($Std_{mam}$), to assess whether there is significant seasonal difference in the IAVs.

At 200 hPa, ozonesonde observations show significantly greater IAV in MAM than DJF over both regions. At 400 hPa and 700 hPa, there is not significant seasonal difference in the magnitude of $O_3$ IAV between MAM and DJF, except for over North America at 400 hPa, where observed $O_3$ IAV is greater in MAM than DJF.

*Question 5, The author's state that the correlation between polar winter 150 mb temps and 200 mb ozone anomalies being lower in spring is "consistent with our understanding of the impact of*

*temperature variations on the formation of polar stratospheric clouds and polar vortex isolation with reduced transport of o3 from the tropics at low temperatures....". I personally don't follow this at all. Are you trying to explain why there is a correlation, or why the correlation is different between winter and spring?*

We deleted our discussion about the relationship between $O_3$ IAV and temperature at 150 hPa averaged over latitude north of 60°N. The averaged temperature is a good measure of the overall temperature in the polar vortex (https://ozonewatch.gsfc.nasa.gov/facts/vortex_NH.html). Although data show the high correlations between polar vortex temperature and $O_3$ IAV over selected sondes station in the lower latitudes, we cannot derive the directly causality without more detailed examinations.

*Question 6, I think you need a quantifiable definition of what you mean by IAV in order to compare where it is larger or smaller in different seasons or in different regions. The paper is written as though IAV is the same as the deviation (anomaly) from the seasonal mean. One then has to determine the interannual variations from looking at wiggles in anomaly plots.*

Please see our response to Question 4.

The definitions of IAV amplitude has been added in text. Text has been modified based on the statistical comparisons of standard deviations.

*Question 7: Discussion of Table 3, Please explain how, from looking at the correlation coefficients in Table 3, that one concludes that 27% of the NA interannual variation is related to 200 mb changes in winter.*

We calculate the percentage of variance explained ($r^2$) through the correlation. The correlation between $O_3$ anomalies at 200 hPa and 400 hPa in winter is 0.52 means $0.52^2 \times 100 = 27\%$ of the variance in 400 hPa is "explained" or related to 200 hPa $O_3$ anomalies.

To avoid confusion, we replaced r with $r^2$ in Table 3 and modified corresponding discussions in the text. We also add the definition of explained variance in the revised manuscript.

*Question 8: Discussion of Figure 6, Mt Pinatubo erupted in June 1991. Your 700 mb DJF NA plot shows a large difference between the red, black and green lines for 1990. What are you defining as the "Pinatubo period" and do you keep 1990 in your re-calculations of strato3-o3 correlation when you say you omit the Pinatubo period?*

We define the Pinatubo period as year 1991-1995. No, the re-calculation is from 1996 to 2016. The text has been modified to avoid the confusion.

*Question 9: around line 260-265 it states that anomalies in strato3 diverge from simulated o3 near the end of the period, and looking at figure 5, that seems to be around 2012. Do precursors really become significantly important only in the past decade?*

No. Precursors are important through the whole time period in the lower troposphere, especially over Europe, where there are less stratospheric intrusions. That is why we see small correlations between $StratO_3$ and $O_3$ at 700 hPa. Below figure shows the $StratO_3/O_3$ averaged over Europe sites

at 700 hPa and 900 hPa in boreal winter season (DJF) from 1990 to 2016. We can see that the StratO$_3$/O$_3$ ratio is less than 0.5 and decreases sharply at 900 hPa after 2014.

[Figure]

Figure R1: Time series of StratO$_3$/O$_3$ averaged over North America sites (top) and Europe sites (bottom) at 700 hPa and 900 hPa in boreal winter season (DJF) from 1990 to 2016.

We modified the text as below:

There is no significant relationship between StratO$_3$ and simulated O$_3$ at 700 hPa. This is expected since the impact of stratospheric ozone decreases, and the impact of ozone production from its precursors becomes more important at lower altitudes.

*Question 10: If you separate the analysis more finely than simply Europe vs NA, and compared comparable latitudes, do you come to the same conclusions? How different are Madrid and Wallops? Your NA comparison includes more high latitude stations than your European one does. Is it longitude you're finding differences between, or latitude?*

Regarding to reviewer's comments about N. American sondes, we have analyzed the latitudinal difference of N. America ozonesonde by separating ozonesonde stations into 3 groups (> 70N, 70N-50N, and <50N). We do find that O$_3$ IAV over N. America varies with latitudes, but the longitudinal difference of StratO$_3$ influence to the troposphere between N. America and Europe is persistent over most NH mid-high latitudes (Figure R2).

We identified that the stronger and deeper stratospheric O$_3$ influence over N. America than Europe through the comparisons sampled at sonde stations. In section 5.2, we extend our analysis from O$_3$ sampled at stations to the latitudinal average between 30°N and 80°N. As shown in Figure 9 and 10 in revised manuscript, the stronger and deeper stratospheric O$_3$ influence over N. America than Europe is a large-scale phenomenon, and not artificially caused by the locations of sondes stations. Below figure (Figure R2, also Figure S3) shows the climatology map of StratO$_3$/O$_3$ at 400 hPa in DJF and MAM averaged from 1990 to 2016. Red thick line is the location of strongest winds, which indicates the approximation of the jet climatology locations. Due to large latitudinal temperature and strong westerly upper level winds, the westerly jet breaks down into large-scale eddies, which are called the baroclinic eddies. The baroclinic eddies push warm air poleward and cold air southward, cooling the subtropics and warming the polar latitudes, in wavelike pattern. As we can see from the figure below, the jet meanders to the south over central and eastern N. America and bringing cold polar air with more stratospheric subsidence. The jet moves to the north over Europe and brings in warm air with less stratospheric O$_3$ influence. The longitudinal difference is persistent between N. America and Europe over most mid-high latitudes.

[Figure]

Figure R2: Spatial maps of simulated StratO$_3$/O$_3$ ratio climatology at 400 hPa in DJF (top) and MAM (bottom) averaged from 1990 to 2016. Red thick lines indicate the approximated climatological jet locations, where the strongest winds are.

*Question 11: On line 306, replace "changes" with "relationship" Your plot shows snapshots of winter and spring 1993, not differences (or changes).*

The text has been modified as suggested.

*Question 12: Final paragraph, the implication here is that the underlying meteorology was deficient over NA in the early period, but perhaps not over Europe. What would be the reason for that? And, can you look at any other fields in the model/sonde comparisons to assess whether this is the issue (maybe tropopause pressure, or the temperature from the radiosonde that flew with the ozone sonde?*

One possible reason is related to the spatial representativeness of meteorological measurements over these two regions. As we can see from Figure 7a in revised manuscript, the westerly subtropical jets show a southward shift over N. America and moves northward over Europe. Most stations over Europe are located south of the subtropical jet, with less dynamic perturbation. Over N. America, the excessive STE are inferred over two stations between 50N and 70N (Figure S2), which are on the edge of the subtropical jet, a region with complex metrological regimes and strong O$_3$ gradient. Considering the much coarser and low-resolution observations in the underlying meteorology in earlier period, problems tends to occurs over a region with more dynamical perturbations (e.g. N. America) than a meteorologically stable region (e.g. Europe).

Evaluating the regional accuracy of underlying meteorology in the early period is beyond the scope of this paper.

*specific comment: please change "amplitude" on line 194 to "magnitude".*

The text has been modified as suggested.
* * *
***Anonymous Referee #2***

*Liu et al. use model simulations of ozone and a stratospheric ozone tracer together with observations from ozonesondes to investigate the interannual variation of ozone and the vertical extent of the impact of stratospheric ozone on tropospheric ozone. Before the simulations are used for the analyses their quality is checked by first comparing the simulations to measurements. I am confident that the study itself is important and deserves to be published, however, I am not happy with how the result from these studies are presented. The manuscript in its present form is confusing and needs thorough structuring and a clear line. From the current manuscript is not clear what the major focus of this study is: Do you want to evaluate the model or do you want to investigate the stratospheric impact on the NH winter and spring interannual variability in the troposphere as it is stated in the title.*

*The manuscript in its current form has a stronger focus on the evaluation of the model than on the analyses of the interannual variability. Further, a lot of information is packed into the figures and thus makes it quite hard to follow and get the major results through. I would suggest major revisions before the manuscript can be published.*

We acknowledge the comments by the second reviewer. But we disagree with the reviewer's comments "The manuscript in its current form has a stronger focus on the evaluation of the model than on the analyses of the interannual variability." The purpose of the paper is not just model validation, but primarily to use observations, model and StratO$_3$ to answer the question of how the stratospheric O$_3$ impacts the troposphere O$_3$ IAV.

Considering that there is no publication on evaluation of the tropospheric O$_3$ simulation from the MERRA2-GMI run, we think it is very important to do the model evaluation before using the model to explain the cause of tropospheric O3 variations.

*Specific comments:*

*P1, general: Why is it important to look at the interannual variation? What are the unanswered questions? The motivation for this study is not clear. In the introduction (P2, 58-59) a motivation is given. Something like this could be repeated in the abstract.*

Using the interannual variation is a good way to evaluate stratospheric impact, since we are looking at the response of tropospheric ozone to stratospheric forcing.

We have a brief discussion of motivation in the 1$^{st}$ paragraph of the introduction, which lead to the main objective of our study in the 2$^{nd}$ paragraph of the introduction. In the 3$^{rd}$ and 4$^{th}$ paragraph we give a more detailed description of background and unanswered questions of this topic and our approach to achieve the goal. We think the motivation are sufficiently described in the introduction and we don't think it is necessary to include it in the abstract.

*P1, L1: How long is the model run? That should be mentioned here.*

The run period has been added in section 2.2. The analysis period is added in the abstract.

*P1, L29-30: Why should ozone sondes be closer to the polar vortex? This sentence is somewhat weird and misleading and thus should be rephrased.*

Discussion has been modified to avoid the confusion.

Please see our response to the Question 1 from the reviewer 1.

*P2, L44: What exactly are these "replay" simulations? This should be explained. What atmospheric conditions or initial conditions have been assumed for this simulation?*

Please see our response to Question 2 from the reviewer 2.

*P2, L48: Which parameters exactly? Can you give some examples?*

We replace 'parameters' into 'system'. The parameters we used in our study include air mass flux, tropopause pressure and geopotential heights.

*P3, L75ff: Here you give a better description of the aim of this study. Something like this should be also added in the abstract, so that it also there becomes more clear why it is important to investigate these processes.*

Please see our response above. We discuss the objective of this study in 2$^{nd}$ paragraph of the introduction.

*P3, Section: A comparison for each station would also be quite useful to understand local differences and which stations/locations maybe mess up the mean.*

Our examination on individual stations shows that the underestimate of tropospheric $O_3$ depletion during the DJF and MAM of the Pinatubo period exist over most N. American stations. Simulations at Wallops Island did a better job among all the N. America stations. Over Wallops Island, model reproduce the $O_3$ variation at 400 hPa, but still underestimate the decreased $O_3$ at 700 hPa.

*P5, L135ff: The comparison to the satellite data has not been mentioned in the abstract or introduction. Why? If it is a part of this study it should be mentioned there. Why do you this comparison in the first place? Is this really necessary? You anyway comparethe model simulations to sonde data so. Therefore, I do not understand what additional information is gained by doing an additional comparison. Especially, if your focus is not on the evaluation of the model but on the investigation of the impact of stratospheric ozone on tropospheric ozone.*

We think it is necessary and important to include satellite comparison. We want to know that the model performs well in the large scale before looking at sondes. But we can put these figures in the supplement if the reviewer insists.

*P5, L154ff: Reference to the figure is missing.*

The reference to the figure is at the end of the sentence.

*P7, L205ff: I cannot follow how you derive this conclusion. Which season and time periods are you referring to? How have the numbers in percent been derived?*

Please see our response to Question 7 of the reviewer 1.

To avoid confusion, we replaced r with $r^2$ in Table 3 and modified corresponding discussions in the text.

*P8, L228: What exactly is the StratO3 tracer? What is included in the diagnostic? How is it calculated? Is this simply the stratospheric O3 flux?*

More detailed discussions of the StratO$_3$ tracer setting in the model have been added in section 2.2. Please see below:

A StratO$_3$ tracer is included in the model to track the stratospheric O$_3$ influence on the troposphere. StratO$_3$ is set equal to simulated O$_3$ in the stratosphere and is removed in the troposphere based on interannually-varying monthly mean loss rates and surface deposition fluxes archived from the standard full chemistry simulation, thus diagnosing the relative importance of stratospheric ozone at all locations in the troposphere. StratO$_3$ tracer is defined relative to a dynamically varying tropopause tracer (e90) (Prather et al., 2011). The e90 tracer is set to a uniform mixing ratio (100 ppb) at the surface with a 90-day e-folding lifetime everywhere in the atmosphere. This lifetime is long enough for the tracer to be well mixed throughout the troposphere but short compared to the transport time scales in the stratosphere, resulting in sharp e90 gradients across the tropopause. In our simulations, the e90 tropopause value is set to 90 ppb. The e90 tracer has been used in many studies of STE as an accurate tropopause definition and an ideal transport tracer in UTLS (e.g., Hsu and Prather, 2014;Liu et al., 2016;Pan et al., 2016;Randel et al., 2016;Liu et al., 2017).

*P8, L234: Where exactly do we see this in Figure 6?*

Figure 5 e, f in the revised manuscript. The reference to figure has been added in the text.

*P9, L266ff: Also here it is not clear how the numbers in percent have been derived.*

Please see our above response. The numbers are square of correlation coefficients.

*P9, L267-267: Here an important result is given, but it gets somehow lost in the discussion of the differences between the model simulations and observations.*

We include this result in the abstract.

*P9, L269: Reference? Has this relation seen before? Has this relation already been discussed somewhere else?*

This sentence provides a hypothesis to explain the difference in the stratospheric $O_3$ influence between North America and Europe as shown in Figure 5 in the revised manuscript. We move this sentence to the next paragraph to lead the discussion of Figure 6 in the revised manuscript.

*P10, L298-299: This sentence is too complicated and should be rephrased. Maybe it would be better to split this sentence also into two sentences.*

The text has been modified as suggested:

In the equatorward breaking, tongues of high PV and stratospheric air extend equatorward associated with frequent STE processes. In the poleward breaking, tongues of low PV and upper tropospheric air extend poleward.

*P10, L308: It would be worth to more clearly state that because of the different tropopause heights different pressure levels are shown in the figures.*

The text has been modified as suggested.

*P10, L315: How it the air mass flux derived/calculated?*

The air mass flux is air mass flow rate, which is calculated by multiplying omega (the volume flow rate which depends on the pressure difference) with the density of air.

The text has been added into revised manuscript.

*P10, L320: not shown? Or is this shown? Can this be seen when comparing 1993 to 1998?*

This can be seen when compare 1993 and 1998. You can see the difference of longitudinal variations of subtropical jets between Figure 7 and 8 in revised manuscript.

*P10, general: In the introductory part of this section StratO3/O3 distinction based on PV is mentioned, but in the analyses the air mass flux is used.*

PV is mentioned in Thorncroft's paper to characterize these wave-breaking events, which also closely associated with STE process. In our analysis, we rely on air mass flux to infer the strength of STE process.

*P11, L327: Here four panels are given, but only 2 panels show the 400 hPa level.*

The labels have been corrected in the text.

*P11, L330: Why is there less dynamic perturbation?*

We infer this from the maps of winds (Figure 7a) and air mass flux (Figure 7c) in the revised manuscript. Both horizontal and vertical transport is smaller over north of 70°N than regions between 50°N and 70°N.

*P12, L363: Why are these three parameters used? What is the connection between these? This is not really discussed. Wouldn't it then be easier to just show StratO3/O3?*

Figure 9 and 10 in the revised manuscript examine whether the longitudinal variations of $StratO_3$ influence on tropospheric $O_3$ inferred from observations and simulations over North America and Europe sonde stations is a large-scale phenomenon and related to the large-scale circulation patterns. The geopotential heights and tropopause pressure are good representors of large-scale circulation patterns. We therefore use these two parameters to represent the dynamic system.

*P12, L383: maximum? Shouldn't it read minimum? Generally, I have the feeling that in this paragraph the description does not agree with the figure shown.*

We change 'correlation' into 'anticorrelation". In this way, it is correct to say the anticorrelation reaches maximum at the surface.

*P13, L396: This is not clear. How does the Pinatubo eruption deplete ozone? Do you mean in the troposphere or the stratosphere and by which process?*

There are many studies examine the stratospheric and tropospheric $O_3$ depletion after the Pinatubo eruption through dynamics and chemistry processes. Please see discussion in section 4.1.

*P13, L410-411: This does not become comprehensible from what is shown in the manuscript.*

This is a conjecture based on our analysis. The observations show that tropospheric $O_3$ decreases after the Pinatubo, reflecting the decreased $O_3$ as seen in the stratosphere. Model reproduced the observed stratospheric $O_3$ decrease, but did fully reproduce the observed tropospheric $O_3$ decrease. Our model analysis shows that there is an increase of $StratO_3$ in the troposphere after the Pinatubo eruption. $StratO_3$ changes in the troposphere are due to two factors: ozone concentrations in the stratosphere, and the mass flux from stratosphere. We therefore speculate that model may overestimate the downward flux at this period and the effect of decreased stratospheric ozone to the tropospheric $O_3$ could thus masked by this overestimation in the model analysis.

*Figure 2 and 3: Are these figures really useful? Especially, since later anyway the simulations are compared to ozone sonde data. This part of the study could (if required) be provided in the supplement.*

We think that all these figures lend credibility to the model. We think it is necessary and important to include satellite comparison. We want to know that the model performs well in the large scale before looking at sondes.

*Figure 4: What does the reader gain from this Figure? Is there any more information gained when comparing observations from all stations with the model simulation?*

We remove the Figure 4 and its discussion as suggested by the reviewer.

*Figure 5, 6, and 7: I would suggest to split these by North America and Europe and discuss the regions separately. As you do it now, you compare different pressure levels, seasons and regions and it gets really hard to follow since you also above all that additionally discuss the differences between model simulation and observations.*

We keep these figures unchanged, since they show direct regional and seasonal comparison.

But we modify the text to discuss the regions separately and a summary of the difference between N. America and Europe.

*Figure 8, 9: Again too many panels and too many things discussed at the same time. I would suggest to solely show the anomalies in the figure and to provide the airmass flux in the supplement.*

There are four situations in flux change: 1) increase of downward flux, 2) decrease of upward flux, 3) decrease of downward flux 4) increase of upward flux around the tropopause. We cannot distinguish these four situations based only on the anomalies of airmass flux. We have to combine the maps of air mass flux and its anomalies to determine how flux changes.

*Technical comments: P2, L18: add "of O3" after input and maybe use a different wording*

*for "input", e.g. entrainment.*

The text has been modified as suggested.

*P2, L47: "in so doing" ! "in doing so"?*

The text has been modified as suggested.

*P4, L99: present = 2019? It would be better to clearly state the year here.*

The text has been modified as suggested.

*P4, Section 4 header: remove colon.*

The text has been modified as suggested.

*P4, Section 4.1 header: remove full stop after title.*

The text has been modified as suggested.

*P7, L219: space between "correlation" and reference of "Terao" missing.*

We have the reference of "Terao et al 2008"

*P7, Section 4.3 header: Remove colon.*

The text has been modified as suggested.

*P12, L360: "impact on tropospheric O3 from the upper to lower troposphere" ! not clear. Please rephrase the sentence.*

We rephase the sentence into: the significant impact of the StratO$_3$ IAV on tropospheric O$_3$ reach to the lower troposphere.

*P13, Section 6 header: remove colon.*

The text has been modified as suggested.

*Figure 8 and 9: Panel labelling with a,b,c: : :.. is missing.*

Labels have been added in the figures.

*Figure 8: Adjust both columns so that they are next to each other at the same height. At the moment there is a shift between the columns.*

Figure 8 has been reproduced as suggested.

*Figure 10 and 11: 180 W on the right side of the x-axes should read 180 E.*

The label has been corrected.

*Figure 10 and 11: To use white dashed lines instead of black dashed lines would increase the readability.*

We tried the white lines. The effect is not good. We therefore keep the black lines.

*Figure 12: Also here North America and Europe should be marked.*

The figure has been modified as suggested.
* * *
***Anonymous Referee #3***

*The paper compares the 1990-2015 ozonesonde observations at 8 North American and 9 European sites with CCM output of tropospheric ozone levels to study the stratospheric impact on the observed tropospheric ozone concentration time series. The (total + tropospheric) ozone output of the model is first validated by comparison with satellite ozone retrievals. Making use of a model stratospheric ozone tracer, the impact of STE on tropospheric ozone is assessed, together with the analysis of model wind patterns and airmass fluxes.*

*GENERAL COMMENTS*

*The study is scientifically sound and takes into account all relevant literature. The analysis is detailed and all relevant aspects are considered. The presentation is clear, although somewhat verbose at some locations, and follows a very logical structure. It therefore deserves publication in ACP, if some remaining issues can be described better or clarified. These are summed up here below.*

Thanks a lot for the comments by the third reviewer.

*SPECIFIC COMMENTS*

*\* From the text (page 4, lines 114-120), it is not clear how the stratospheric ozone tracer (StratO3) is defined. Please be more specific on this important variable of your analysis.*

More detailed discussions of the StratO$_3$ tracer setting in the model have been added in section 2.2.

*On page 5, lines 147-150: please, be more quantitative when comparing the magnitude, IAV and trend of the tropospheric ozone satellite retrieval and model replay simulation.*

*More in general, I agree with reviewer 1 that, throughout the entire manuscript, you should quantify the comparison of "IAV" between two datasets.*

Please see the response to Question 4 of the reviewer 1.

The revised manuscript included two statistical tables as supplementary materials. Those tables include 1) standard deviations of each time series (representing of IAV), as well as F-test statistics to assess whether there is a significant difference in the IAVs 1) between North America and Europe, 2) between DJF and MAM. The corresponding discussion are added into text along the lines we discussed about seasonal and regional IAV.

*On Page 5, lines 154-156, please describe more clearly how the ozone anomalies are calculated. For instance, for every ozonesonde site, you first calculate the monthly anomalies, and then you calculate the monthly mean of those monthly anomalies for all sites together? What does the 95% confidence interval represents ? The site to site variability with or without the variability within one month at a given site?*

Please see the response to Question 2 of the reviewer 1.

*Coming back to the previous point: quantify the statements on page 6, lines 168-169: "Both observations and simulations show the largest interannual variations in the winter and spring, when the strongest IAVs occur" and on page 6, lines 176-177: "The IAV of ozone is larger over North America then over Europe, and larger in spring than in winter".*

Discussions on IAV comparison have been revised based on statistical analysis.

*In sect 4.1, in which you describe Fig. 4, it should be mentioned that the comparison between ozonesonde data and model simulation decrease with increasing pressure and why this is the case.*

We remove the Figure 4 and corresponding discussion.

*Page 6, lines 184-188: I do not understand the link between the winter polar 150 hPaAVERAGED temperature – 200 hPa O3 IAV correlation and PSC formation, which only happens at very low stratospheric temperatures (< -80_C).*

Please see our response to Question 5 of the reviewer 1.

We deleted our discussion about the relationship between O$_3$ IAV and temperature at 150 hPa average over latitude north of 60°N. Although they show high correlations, we cannot derive the directly causality without more detailed examinations.

*\* Page 7, lines 206-209: where do these explained variances come from (in Table 3, only correlations are shown)? Please explain. Same comment for the percentages for the explained variations, mentioned on Page 8, line 234, and page 9, lines 265-267.*

Please see our response to Question 7 of the reviewer 1.

To avoid confusion, we replaced r with $r^2$ in Table 3 and modified corresponding discussions in the text. We also add the definition of explained variance in the revised manuscript.

*\* Page 9: why are you using the alternative definition of tropopause pressure by Browell et al. (1996)? Is this tropopause identical to the ozonopause? What is the effect of this choice for the tropopause (compared to the thermal tropopause, as defined by the WMO) on the mentioned correlations with the IAV of O3 and stratO3?*

Bethan et al. (1996) has compare the calculated tropopauses using WMO temperature lapse rate criteria with that defined by the ozone gradient. They demonstrated that it is feasible to define the tropopause in terms of ozone concentration, by identifying the sharp gradient in concentration that occurs at the base of the stratosphere. They also argued that for high latitude in winter, by nearly isothermal profile that could lead to indefinite thermal tropopauses. Another reason is that for ozonesonde data, we did not process its co-measured temperature profile. We therefore used ozone tropopause here.

*\* Page 12, lines 372-388: the analysis of the correlations between AO and ozone is not very convincing. First of all, please mention the months for which Fig. 12 is constructed (DJF and/or MAM?). Secondly, on which ground do you classify the correlation profiles (with low correlation coefficients after all) in Fig. 12 as significantly different between North America and Europe? And similar between sonde and model data in Figure S2?*

Figure R3 (Figure 11 in the revised manuscript) shows the correlation during the winter season. The caption has been modified. We are arguing the deeper and stronger AO-$O_3$ coupling over N. America than over Europe. We add dashed lines to indicate regions with statistically significant correlation (df=25, p<0.05). Please see below figure.

[Figure]

Figure R3: Longitudinal variations of correlation profiles (r) between AO index and simulated $O_3$ averaged over 30°N and 80°N in DJF from 1000 hPa to 200 hPa. Correlations inside black dashed lines are statistically significant (df=25, p<0.05). Red dashed lines indicate the longitudinal range for the North American region (120°W-60°W) and the European region (10°W-26°E).

*TECHNICAL CORRECTIONS*

*\* Pag 1, line 29: remove the ' after ozonesondes*

The text have been modified as suggested.

*\* Page 2, line 46-47: replace "In so doing" with "In doing so".*

The text have been modified as suggested.

*\* Page 5, before Section 4: Here, you can add that some features in tropospheric ozone are well reproduced (e.g. 2015), while others not (e.g. 2013) and that those differences will be analyzed further in the paper.*

The text has been modified as suggested.

*\* Page 9, after line 269: please mention here that the longitudinal difference in dynamics between North America and Europe will be further analyzed in Sect. 5.2.*

We move this sentence to the next paragraph to lead the discussion of Figure 6 in the revised manuscript.

*\* Page 10, line 310: replace "asterisks" by "lines" (referring to Fig. 8)*

The text has been modified as suggested.

*\* Page 13, line401: replace "resulting" with "result".*

The text has been modified as suggested.

*\* Please remove the : in the section titles (e.g. 6: Conclusions and discussion)*

We removed all the : in the section titles.

*\* Please acknowledge the data repositories properly for the ozone data used (ozonesondes: WOUDC, SBUV, OMI, etc.).*

Below texts are added in the acknowledge:

We thank the World Ozone Data Centre and the SHADOZ program for making the routine sonde data accessible. We gratefully acknowledge Dr. Jerry R Ziemke from NASA for providing the OMI/MLS TCO data and Dr. Stacey M. Frith from NASA for providing SBUV total ozone column data. We thank the reviewers for their helpful comments and suggestions to improve this paper.
* * *
**Interactive comment by Ryan Williams**

*r.s.williams@pgr.reading.ac.uk*

*This is an interesting new article on the role of the of the stratosphere on tropospheric ozone interannual variability during Northern Hemisphere winter and spring (when the STE flux is at a maximum). We however feel that our most recent study that looks at the stratospheric influence on tropospheric ozone should additionally be cited within the introduction:*

*"Characterising the seasonal and geographical variability in tropospheric ozone, stratospheric influence and recent changes" by Ryan S. Williams et al. (2019) ([https://www.atmos-chem-phys.net/19/3589/2019/](https://www.atmos-chem-phys.net/19/3589/2019/))*

*We would suggest adding a citation to this paper either on P2, L38: "Stratosphere-troposphere exchange (STE) has been shown to impact the tropospheric ozone distribution (e.g., Terao et al., 2008; Hess et al., 2015; Holton et al., 1995)."*

*Or alternatively on P2, L50: "STE has been widely studied for several decades (Danielsen, 1968; Holton et al.,1995; Olsen et al., 2002; 2003; 2013; Stohl et al., 2003a; 2003b; Sprenger and Wernli, 2003; Thompson et al., 2007; Lefohn et al., 2011; Skerlak et al., 2014)".*

*Since our study does not look at STE explicitly (only implicitly using tagged stratospheric ozone tracers from the EMAC and CMAM CCMs), a citation on L38 would be more applicable in our view.*

Thanks a lot for the short comments on the references. The reference has been added in the revised manuscript as suggested.

*Furthermore, we feel that a mention to nudged, specified-dynamics CCM simulations should be later included in the introduction, in addition to free-running CCM simulations and CTMs (P2-3, L61-72), as a useful tool for assessment of the stratospheric influence on tropospheric ozone (using stratospheric tagged ozone tracers). Compared with free-running CCMs, "the influence on composition of dynamical biases and differences in variability between the reanalysis and the models can be assessed" – Morgenstern et al. (2017), P648 (https://www.geosci-model-dev.net/10/639/2017/). This point could also be made in highlighting the role of constraining the dynamics on influencing the distribution of model composition fields.*

Below text is added in the introduction:

Williams et al (2019) used nudged CCM simulations by the ERA-Interim reanalysis dataset and a stratospheric tagged $O_3$ tracer to assess the role of stratospheric ozone in influencing both regional and seasonal variations of tropospheric $O_3$. Their study shows that stratosphere has a much larger influence than previously thought, although some differences result from the definition of stratospheric tracer.

**Reference:**

Bethan, S., Vaughan, G., and Reid, S. J.: A comparison of ozone and thermal tropopause heights and the impact of tropopause definition on quantifying the ozone content of the troposphere, Quarterly Journal of the Royal Meteorological Society, 122, 929-944, 10.1002/qj.49712253207, 1996.

Chin, M., Ginoux, P., Kinne, S., Torres, O., Holben, B. N., Duncan, B. N., Martin, R. V., Logan, J. A., Higurashi, A., and Nakajima, T.: Tropospheric aerosol optical thickness from the GOCART model and comparisons with satellite and Sun photometer measurements, Journal of the Atmospheric Sciences, 59, 461-483, 10.1175/1520-0469(2002)059<0461:taotft>2.0.co;2, 2002.

Colarco, P., da Silva, A., Chin, M., and Diehl, T.: Online simulations of global aerosol distributions in the NASA GEOS-4 model and comparisons to satellite and ground-based aerosol optical depth, Journal of Geophysical Research-Atmospheres, 115, 10.1029/2009jd012820, 2010.

Duncan, B. N., Logan, J. A., Bey, I., Megretskaia, I. A., Yantosca, R. M., Novelli, P. C., Jones, N. B., and Rinsland, C. P.: Global budget of CO, 1988-1997: Source estimates and validation with a global model, Journal of Geophysical Research-Atmospheres, 112, 10.1029/2007jd008459, 2007.

Hsu, J. N., and Prather, M. J.: Is the residual vertical velocity a good proxy for stratosphere-troposphere exchange of ozone?, Geophysical Research Letters, 41, 9024-9032, 10.1002/2014gl061994, 2014.

Liu, J., Rodriguez, J. M., Thompson, A. M., Logan, J. A., Douglass, A. R., Olsen, M. A., Steenrod, S. D., and Posny, F.: Origins of tropospheric ozone interannual variation over Reunion: A model investigation, Journal of Geophysical Research-Atmospheres, 121, 521-537, 10.1002/2015jd023981, 2016.

Liu, J., Rodriguez, J. M., Steenrod, S. D., Douglass, A. R., Logan, J. A., Olsen, M. A., Wargan, K., and Ziemke, J. R.: Causes of interannual variability over the southern hemispheric tropospheric ozone maximum, Atmospheric Chemistry and Physics, 17, 3279-3299, 10.5194/acp-17-3279-2017, 2017.

Orbe, C., Oman, L. D., Strahan, S. E., Waugh, D. W., Pawson, S., Takacs, L. L., and Molod, A. M.: Large-Scale Atmospheric Transport in GEOS Replay Simulations, Journal of Advances in Modeling Earth Systems, 9, 2545-2560, 10.1002/2017ms001053, 2017.

Pan, L. L., Honomichl, S. B., Kinnison, D. E., Abalos, M., Randel, W. J., Bergman, J. W., and Bian, J.: Transport of chemical tracers from the boundary layer to stratosphere associated with the dynamics of the Asian summer monsoon, Journal of Geophysical Research-Atmospheres, 121, 14159-14174, 10.1002/2016jd025616, 2016.

Prather, M. J., Zhu, X., Tang, Q., Hsu, J. N., and Neu, J. L.: An atmospheric chemist in search of the tropopause, J GEOPHYS RES-ATMOS, 116, D04306, 10.1029/2010jd014939, 2011.

Randel, W. J., Rivoire, L., Pan, L. L., and Honomichl, S. B.: Dry layers in the tropical troposphere observed during CONTRAST and global behavior from GFS analyses, Journal of Geophysical Research-Atmospheres, 121, 14142-14158, 10.1002/2016jd025841, 2016.

Strahan, S. E., Duncan, B. N., and Hoor, P.: Observationally derived transport diagnostics for the lowermost stratosphere and their application to the GMI chemistry and transport model, Atmospheric Chemistry and Physics, 7, 2435-2445, 2007.

Williams, R. S., Hegglin, M. I., Kerridge, B. J., Jockel, P., Latter, B. G., and Plummer, D. A.: Characterising the seasonal and geographical variability in tropospheric ozone, stratospheric influence and recent changes, Atmospheric Chemistry and Physics, 19, 3589-3620, 10.5194/acp-19-3589-2019, 2019.

---

## Author Response (AR3)

**2nd Response to reviews on "Stratospheric impact on the Northern Hemisphere winter and spring ozone interannual variability in the troposphere"**

**by Junhua Liu et al.**

We thank the two reviewers and the editor for their helpful comments. We have addressed all comments in detail below and have clarified the text in the relevant sections.

In the following, we address the concerns raised by all the reviewers. Reviewers' comments are italicized.

Reviewer 1:

1) *Please include upfront a clear definition including one that is mathematical of what you mean by IAV*
   The definition of IAV is the year by year variations in the time series. The magnitude of IAV is resented by its standard deviation. The equation of calculating anomalies has been added into supplement.

2) *The paper needs to be grammar checked; in particular pay close attention to verb tense.*
   Grammar check has been done by the second and first authors.

3) *When introducing the model runs on line 77, make sure you note clearly this a specified-dynamics, or replay or nudged run*
   Text has been modified as suggested.

4) *line 98/99 related to #3, Davis et al just published a paper detailing issues regarding STE and nudged climate model runs. (see DOI 10.5194/gmd-13-717-2020). Bottom line is that with at least some nudging techniques the vertical transport that is the input (ie, like the MERRA2 winds) is not what you get after the nudging occurs, and it can give incorrect trends. A comparison of what you call IAV in a free running model with what you have in the replay version (of course using the same chemistry) would be useful. This may be beyond the scope of this study, but could impact some of your results. I would strongly encourage the authors to consult Dr. Orbe (who's paper they cite) as to possible deficiencies in the results presented here.*
   We have consulted with Dr. Orbe and added corresponding discussion at the end of section 5.
   Apart from the input meteorological fields, the discrepancies might be also due to the replay configuration used in the model. Orbe et al (2016) showed that small differences are seen in stratosphere-troposphere exchange between the GMI-CTM and a replay simulation constrained with the same meteorological fields. In spite of the weaker response in the model, the general agreement between the model and observations and the correlation between $StratO_3$ and measurements indicate a significant impact of stratospheric ozone variations on tropospheric ozone.

   Below are more detailed explanations.

Orbe et al (2017b) compared the transport characteristics of several different models that use the same meteorological fields taken from MERRA and found that the NASA GEOS-replay tends to simulate greater downward flux over the NH troposphere during winter, compared to the GMI chemistry transport model (Figure 2 of Orbe et al. 2017b). But the bottom line is that we don't know which one is right. The replay configuration in Orbe et al. (2017b) is slightly different to that in our model, another study by Orbe et al (2017a) found that negligible differences in stratosphere-troposphere exchange between these replay configurations (Figure 8 and its discussion of Orbe et al. 2017a). Overall, the conclusion is STE in the model shows a weak response to the different model configurations (CTM vs replay and different replay configurations).

5)  *StratO3: The description of the quantity needs to be improved. I don't understand how you are tagging stratospheric ozone and I also would like to see some statement as to how using E90 as a tropopause definition compares to using an actual tropopause definition (such as PV or the vertical temperature gradient).*
Below texts are added in the paper:
The e90 tropopause is optimal in effectively separating stratospheric from tropospheric air from a chemical composition perspective compared to other traditional definitions of the tropopause (Prather et al. 2011), and offers the advantage of being able to be calculated online in the model.

6)   *line 131 (section 3); One of the other reviewers commented that there seems to be two topics, model validation and then the tropospheric ozone analysis. Is the issue that this way of running the model has not been validated with respect to tropospheric ozone? Perhaps there should also be a model validation paper, or if not, then this section might be better as supplementary material or as an appendix, because it distracts from the main science being presented.*

We deleted this section and put the two figures in this section into supplementary materials. We added below sentence at the end of second paragraph of section 2.2:

Our initial model evaluations suggest that the MERRA2-GMI replay ozone simulation are in good agreement with the seasonality and IAV of the total and tropospheric column ozone from satellite observations (Figure S1 and S2).

7)  *line 167 and subsequent paragraph: What is impact? It needs some reworking. And, also on the line, are you referring to the mean flux of ozone from stratosphere to troposphere (or mass flux...ie see Appenzellar et al, 1996, JGR, 10.1029/96JD00821).*

Texts have been modified as suggested.
Previous studies have shown that the relative contribution of stratospheric ozone to tropospheric ozone is greatest in the free troposphere during winter (e.g., Holton et al., 1995;Stohl et al., 2000;Skerlak et al., 2014;2015) and at the surface during spring (e.g., Lin et al., 2012;2015). In summer, the relative contribution of stratospheric ozone is low due to the increased chemical ozone production in the troposphere.

8)  *Some of the terminology is kind of confusing...specifically when you talk about explaining the variance of the IAV (which is already a variation). I think you're explaining the variance of ozone, not the ozone IAV. Giving a mathematical definition of IAV may help.*

In statistics, explained variance ($r^2$) measures the proportion to which a data set (e.g. ozone anomalies at 200 hPa) accounts for the variation of a given data set (e.g. ozone anomalies at 400 hPa).
Here we are using the statistic term "explained variance" to determine the fraction of the ozone variance that in the troposphere that can be attributed to the variance in stratospheric ozone. For example, if explained variance eq 1, that means 100% of ozone variations at 400 hPa is explained from ozone variations at 200 hPa.

The text has been modified to avoid the confusion.

9)  *The paper really needs to have the language cleaned up in regards to the way it talks about variations and variance and changes. One example, the text states on line 232/233 "The IAV of StratO3 tracer in the troposphere reflects a combined effect of the changes in the lower stratospheric O3 concentrations and in the strength of stratosphere-to-troposphere (STE) mass flux." I think what you're saying is that variability in the amount of tropospheric ozone that was transported from the stratosphere is due to both variability in the lower stratospheric ozone reservoir and variability in the net downward mass flux. (and Albers et al, JGR, 2018, doi 10.1002/2017JD026890 would be a reasonable reference for that).*

Text has been modified as suggested.
The variability in the amount of tropospheric ozone that was transported from the stratosphere as inferred by $StratO_3$ is due to both the variability in the lower stratospheric ozone reservoir and the variability in the net downward mass flux (Albers et al., 2018). These two variabilities may either cancel or reinforce each other, depending on their relative phases.

10) *One comment: Line 271/272 says " The difference in the stratospheric O3 influence between North America and Europe is likely due to longitudinal difference in dynamics." I am not sure what that actually means. Perhaps better would be to say it is due simply to the existence of planetary scale waves (which are part of dynamics). I have a similar comment regarding line that states " Figure 7 illustrates the relationship of StratO3/O3 ratio at 400 hPa to dynamics ..." You are not illustrating a relationship to dynamics, but to specific quantities.*

Texts has been modified as suggested.
Line 295/296 in revised manuscript with tracked changes:  The differences in the stratospheric ozone impact between North America and Europe are likely due to variations in the net downward flux associated with planetary-scale waves.

Line 313: Figure 5 illustrates the relationship of $StratO_3/O_3$ ratio at 400 hPa to horizontal winds at 400 hPa, and vertical air mass flux near the seasonal mean tropopause pressure in the year 1993.

11) *This is another sentence that is confusing: "The vertical airmass flux is calculated by multiplying w (the volume flow rate, which depends on the pressure difference) with the density of air. " Isn't omega dp/dt? (I'm not sure what a volume flow rate means here).*
Yes. It is omega, depends on the pressure difference dp/dt.
The vertical airmass flux is calculated by multiplying omega (dp/dt, units: pa/s) with density of air. The sign of calculated air mass flux is reversed so that positive values are upward fluxes, negative values are downward fluxes.

12) *And, I would just refer to it as the downward mass flux (rather than vertical airmass flux).*
It would be confusing to replace 'vertical mass flux" with "downward mass flux'. In the plots, positive values are upward fluxes, negative values are downward fluxes. I used google scholar to search 'vertical air mass flux' and 'downward air mass flux'. The first term gave back more results than the second term.

13) *And, this is another confusing statement " The jet locations, approximated by the strongest winds". Jet locations are where the strongest winds are, they're not approximated. The rest of this paragraph (centered around line 315) needs to be rewritten. Make sure that all of the co-authors concur with these descriptions (Several of the paper's co-authors should be able to help make this much more understandable. They should also be able to help with the grammar issues that perhaps will also be dealt with by a copy editor.)*

Text has been modified as suggested:
The jet, the location of maximum winds, are indicated by red thick lines.

We rewrote this section (4.1) to make the discussion clear and precise.

14) *lines 358/259 states " Our analysis suggests that the IAV of wave disturbances in the westerlies likely affect the IAV of the regional distributions of prevailing wind patterns as well as the strength of STE flux. " Wave disturbances (or better, just waves) and wind are going to co-vary, and the location where tropopause folds occur will co vary with the waves too. This whole discussion needs to be rewritten.*

We rewrote this discussion as below:
Our analysis suggests that the IAV of wave disturbances in the westerlies likely affects the IAV of the regional distributions of prevailing wind patterns as well as the strength of downward airmass flux across the tropopause. The IAV of the stratospheric ozone influence in the troposphere reflects a combined effect of the changes in the lower stratospheric ozone concentration and in the 3-d dynamics, which may either cancel or reinforce each other.

*15) The overall results of this study seem to be*
*a) from the first part, the model seems to do OK at reproducing means and variability as based on comparisons with SBUV and sondes (and this should be an appendix or supplement.)*
*b) temporal variability in tropospheric ozone in the NH mid latitudes is related to temporal variability in the transport of ozone from the stratosphere*
*c) there is a difference in the degree of influence of stratospheric ozone on tropospheric ozone between North America and Europe. This is essentially a function of the spatial distribution of regions of tropopause folding type events, which is really a consequence of the different synoptic patterns the occur over Europe and North America.*
*None of this is really new information, but the particular analysis they do with the stratospheric tracer is fine to publish. They acknowledge a significant number of past similar studies that were done with global scale models. Although they do reference the Lin et al. CalNex paper (however, they do need to fix the citation, the paper was published several years ago), they do not acknowledge that the western US air quality issue related to STE has been studied extensively with aircraft and ground fieldwork, and I would encourage the authors to take a look at some of the papers from the LVOS experiment led by Andrew Langford.*

The two papers by Langford et al (2012;2015) have been added into references for deep STE discussion.

*16) General impression; The authors need to reread all the original reviewer's comments, and respond to those comments in a more thorough manner.*

We believe that in the revised manuscript we have addressed all the comments by the reviewer.

**Reviewer 2:**
*I would like to thank the authors for taking my comments into account and revising the manuscript accordingly. However, one of my comments did not came across well. I did not mean that the model evaluation is unnecessary. My point was that the weighting how the results were described was not done in the correct way. There was to much focus on the evaluation instead of the investigation of the IAV which was supposed to be the main focus of the paper. However, with the revisions done the manuscript reads now much better and I do not feel any mismatch any longer.*

Per the request of the reviewer 1, we moved the results of model evaluation from satellite observations into the supplement.

*Nevertheless, I have several technical comments that should be considered before publication:*

*P2, L45: "replay simulations" → I would suggest to either skip here "replay" or you explain what is meant with replay simulations. Alternatively, you could also add the reference to the Orbe et al. 2017 paper if you do not want to skip replay.*
We skipped here 'replay'.

*P2, L63: I would suggest to skip "tropospheric" since this is confusing. Usually, CTMs simulate the entire atmosphere. If you want to emphasize that the simulation run had a focus on the troposphere it should be said more clearly.*
Text has been modified as suggested.

*P2, L73: Plural or singular? Thus, either "used a nudged CCM simulation" or " used nudged CCM simulations".*
Text has been modified as suggested: "used a nudged CCM simulation"

*P3, L101: Sentence "More detailed information on replay methodology can be found in Orbe et al. (2917)" is obsolete since you refer to the replay simulation and the Orbe et al. reference already a few sentences before.*
We deleted this sentence as suggested.

*P4, L127: in UTLS → in the UTLS*
Text has been modified as suggested.

*P4, L150: Centers → Center*
*P4, L159: Model did not → The model did not*
The section contains above two corrections has been deleted in the revised manuscript.

*P5, L168: relatively weaker → I would suggest to either write "relatively weak" or "weaker".*
Text has been modified as:
"the relative contribution of stratospheric ozone is low due to"

*P5, L171: observed ozone → specify which observations*
We change the text into "ozonesonde measured ozone"

*P5, L183: lost parentheses at the end of the sentence.*
Text has been modified as suggested.

*P5, L203: .....IAV is mall, only become significant.....→ sentence not clear, please rephrase*
We change the text into:
The magnitude of ozone anomalies doesn't show significant seasonal difference between DJF and MAM, except over North America at 400 hPa (Table S2).

*P7, L275: by Browell et al. (Browell et al., 1996) → by Browell et al. (1996)*
Text has been modified as suggested.

*P9, L350: shows the → shows a*
Text has been modified as suggested.

*P9, L355: weakend descending air → please rephrase, e. g. to "weak descending air"*
Text has been modified as suggested.

*P10, L376: N. America → North America*
 Text has been modified as suggested.

*P10, L394: near surface → near the surface*
Text has been modified as suggested.

*P10, L398: show a similar → shows a similar*
 Text has been modified as suggested.

*P10, L403: tropospheric O3 in the troposphere → repetition, please rephrase*
Text has been modified as suggested.

*P15, Table2: Ny Aleasund → Ny Alesund*
Text has been corrected.

*P16, Figure 1: Ny Aleasund → Ny Alesund*
 Text has been corrected.

*P18, Figure 4 and Figure 5: Figures should be done in a consistent way. Here, the x-axis labeling is missing (add year) and the headers should be written in the same way (same font, same abbreviation for North America).*
*P19, Figure 6: Same here as for Figures 4 and 5. Additionally, remove the y-axis labeling in the middle of the figures. N. Am should be N. American toe be consistent and height of the right 4 figures should be adjusted to the left 4 figures.*
*P19, Figure 6 caption: top → Top, bottom → Bottom*

Those three figures (Figure 2-4 in the revised manuscript) have been redone in a consistent way as suggested by the reviewer. The caption has been modified as suggested.

*P20 and 21, Figures 7 and 8: Increase space between the panels. What is the additional colour bar (light pink) for? Cannot this be skipped?*

These two figures (Figure 5-6 in the revised manuscript) have been redone as suggested by the reviewer.

*P20, Figure 7 caption: Do not start sentence with a number → sentence should be rephrased.*
 The caption has been modified as suggested.

Reference:

[revised manuscript text omitted]

---

## Author Response (AR4)

**3rd Response to reviews on "Stratospheric impact on the Northern Hemisphere winter and spring ozone interannual variability in the troposphere"**

**by Junhua Liu et al.**

We have addressed all the points raised in the review and are thankful for the editor and reviewers' suggestions, which contributed positively to the quality of the study.

In the following, we have added point-by-point responses to the editor and reviewers' comments Both the editor and reviewers' comments are italicized.

Editor's comments:
*Please indicate clearly in your answer to the reviewer comments how the comments have been taken into account in the way like this, so that it is possible to check the validity of these changes:*
*on line XXX in the track changed version of the manuscript we have added the following text: " …. ".*
*now for example in answer to reviewer #1 question 10 you give reference to line 315 (which refers to the manuscript without track change) and to line 295/296 (in the track changed version). This is extremely difficult to follow. In other case you simply state that the text has been changed, without explaining how or referencing the line in the (track-changed) manuscript.*
*Please provide this in a way that it can be checked and assessed.*

*With respect to esctino 4.1.: please provide a version which shows the specific changes, and not a version declaring everything as changed.*

Many thanks for the editor's suggestion and I apologize for the confusion.
In the following point-by-point response, we have added the revised text and its referencing line numbers in the track changes version of the manuscript. We have modified the section 4.1 to show the specific changes.

Reviewer 1:

1) *Please include upfront a clear definition including one that is mathematical of what you mean by IAV*

   On line 190 in the track changes version of the manuscript we have added the following text: " To quantify the magnitude of IAVs, we adopt the standard deviation (SD) of these ozone anomalies. We perform the standard statistical F-test to assess the regional and seasonal differences in the ozone IAVs. The calculated standard deviations and F-test statistics are shown in Table S1 and S2.".
   The equation of calculating anomalies has been added into the supplement (Line 14-33).

2) *The paper needs to be grammar checked; in particular pay close attention to verb tense.*
   The grammatical, linguistic errors in the manuscript have been carefully checked and corrected by the second and first author.

We have modified several places in the abstract and throughout the manuscript to clarify the presentation. We tracked all the changes in our manuscript.

3) *When introducing the model runs on line 77, make sure you note clearly this a specified-dynamics, or replay or nudged run*
On line 85 in the track changes version of the manuscript we have added "GEOS-CCM replay" when introducing the simulation. We have also made two minor modifications to this sentence as shown on line 85 and 86 in the track changes version of the manuscript.

4) *line 98/99 related to #3, Davis et al just published a paper detailing issues regarding STE and nudged climate model runs. (see DOI 10.5194/gmd-13-717-2020). Bottom line is that with at least some nudging techniques the vertical transport that is the input (ie, like the MERRA2 winds) is not what you get after the nudging occurs, and it can give incorrect trends. A comparison of what you call IAV in a free running model with what you have in the replay version (of course using the same chemistry) would be useful. This may be beyond the scope of this study, but could impact some of your results. I would strongly encourage the authors to consult Dr. Orbe (who's paper they cite) as to possible deficiencies in the results presented here.*

We have consulted with Dr. Orbe and added corresponding discussion on line 496 in the track changes version of the manuscript: "Apart from the input meteorological fields, the discrepancies might be also due to the replay configuration used in the model. Orbe et al (2017b) showed that small differences are seen in stratosphere-troposphere exchange between the GMI-CTM and a replay simulation constrained with the same meteorological fields. In spite of the weaker response in the model, the general agreement between the model and observations and the correlation between $StratO_3$ and measurements indicate a significant impact of stratospheric ozone variations on tropospheric ozone."

Below are more detailed explanations.
Orbe et al (2017b) compared the transport characteristics of several different models that use the same meteorological fields taken from MERRA and found that the NASA GEOS-replay tends to simulate greater downward flux over the NH troposphere during winter, compared to the GMI chemistry transport model (Figure 2 of Orbe et al. 2017b). But the bottom line is that we don't know which one is right. The replay configuration in Orbe et al. (2017b) is slightly different to that in our model. Another study by Orbe et al (2017a) found that negligible differences in stratosphere-troposphere exchange between these replay configurations (Figure 8 and its discussion of Orbe et al. 2017a). Overall, the conclusion is STE in the model shows a weak response to the different model configurations (CTM vs replay and different replay configurations).

5) *StratO3: The description of the quantity needs to be improved. I don't understand how you are tagging stratospheric ozone and I also would like to see some statement as to how using E90 as a tropopause definition compares to using an actual tropopause definition (such as PV or the vertical temperature gradient).*

On line 132 in the track changes version of the manuscript we have added the following text: " StratO$_3$ tracer is defined relative to a dynamically varying tropopause, which is derived from an artificial tracer, e90, introduced by Prather et al. (2011).".

On line 136 in the track changes version of the manuscript we have added the following text: " The e90 tropopause is optimal in effectively separating stratospheric from tropospheric air from a chemical composition perspective compared to other traditional definitions of the tropopause (Prather et al. 2011), and offers the advantage of being able to be calculated online in the model."

We have also made several minor changes throughout this paragraph to clarify the description.

6) *line 131 (section 3); One of the other reviewers commented that there seems to be two topics, model validation and then the tropospheric ozone analysis. Is the issue that this way of running the model has not been validated with respect to tropospheric ozone? Perhaps there should also be a model validation paper, or if not, then this section might be better as supplementary material or as an appendix, because it distracts from the main science being presented.*

We have deleted this section and put the two figures in this section into the supplement.

On line 126 in the track changes version of the manuscript, we have added the following text to summarize this section: "Our initial model evaluations suggest that the MERRA2-GMI replay ozone simulation are in good agreement with the seasonality and IAV of the total and tropospheric column ozone from satellite observations (Figure S1 and S2)".

7) *line 167 and subsequent paragraph: What is impact? It needs some reworking. And, also on the line, are you referring to the mean flux of ozone from stratosphere to troposphere (or mass flux...ie see Appenzellar et al, 1996, JGR, 10.1029/96JD00821).*

On line 180 in the track changes version of the manuscript, we have modified the text as suggested to: "Previous studies have shown that the relative contribution of stratospheric ozone to tropospheric ozone is greatest in the free troposphere during winter (e.g., Holton et al., 1995;Stohl et al., 2000;Skerlak et al., 2014;2015) and at the surface during spring (e.g., Lin et al., 2012;2015). In summer, the relative contribution of stratospheric ozone is low due to the increased chemical ozone production in the troposphere."

8) *Some of the terminology is kind of confusing...specifically when you talk about explaining the variance of the IAV (which is already a variation). I think you're explaining the variance of ozone, not the ozone IAV. Giving a mathematical definition of IAV may help.*

In statistics, explained variance ($r^2$) measures the proportion to which a data set (e.g. ozone anomalies at 200 hPa) accounts for the variation of a given data set (e.g. ozone

anomalies at 400 hPa). Here we are using the statistic term "explained variance" to determine the fraction of the ozone variance that in the troposphere that can be attributed to the variance in stratospheric ozone. For example, if explained variance eq 1, that means 100% of ozone variations at 400 hPa is explained from ozone variations at 200 hPa.

On line 223 in the track changes version of the manuscript, we have modified the text to: "We use explained variance (square of correlation coefficient: $r^2$) to determine the fraction of the ozone variance that in the troposphere that can be attributed to the variance in stratospheric ozone."
We have also made several minor changes throughout this paragraph to clarify the description.

9) *The paper really needs to have the language cleaned up in regards to the way it talks about variations and variance and changes. One example, the text states on line 232/233 "The IAV of StratO3 tracer in the troposphere reflects a combined effect of the changes in the lower stratospheric O3 concentrations and in the strength of stratosphere-to-troposphere (STE) mass flux." I think what you're saying is that variability in the amount of tropospheric ozone that was transported from the stratosphere is due to both variability in the lower stratospheric ozone reservoir and variability in the net downward mass flux. (and Albers et al, JGR, 2018, doi 10.1002/2017JD026890 would be a reasonable reference for that).*

One line 253 in the track changes version of the manuscript, we have modified the text as suggested: "The variability in the amount of tropospheric ozone that was transported from the stratosphere as inferred by $StratO_3$ is due to both the variability in the lower stratospheric ozone reservoir and the variability in the net downward mass flux (Albers et al., 2018). These two variabilities may either cancel or reinforce each other, depending on their relative phases."

10) *One comment: Line 271/272 says " The difference in the stratospheric O3 influence between North America and Europe is likely due to longitudinal difference in dynamics." I am not sure what that actually means. Perhaps better would be to say it is due simply to the existence of planetary scale waves (which are part of dynamics). I have a similar comment regarding line that states " Figure 7 illustrates the relationship of StratO3/O3 ratio at 400 hPa to dynamics ..." You are not illustrating a relationship to dynamics, but to specific quantities.*

On line 296 in the track changes version of the manuscript, we have modified the text as follows: "The differences in the stratospheric ozone impact between North America and Europe are likely due to variations in the net downward flux associated with planetary-scale waves.".
On line 340, we have modified the text as follows: "Figure 5 illustrates the relationship of the $StratO_3/O_3$ ratio at 400 hPa to the horizontal winds at 400 hPa, and the vertical air mass flux near the seasonal mean tropopause pressure in the year 1993."

11) *This is another sentence that is confusing: "The vertical airmass flux is calculated by multiplying w (the volume flow rate, which depends on the pressure difference) with the density of air. " Isn't omega dp/dt? (I'm not sure what a volume flow rate means here).* Yes. It is omega, depends on the pressure difference dp/dt.

On line 344 in the track changes version of the manuscript, we have modified the text as follows: "The vertical airmass flux is calculated by multiplying omega (dp/dt, units: pa/s) with density of air. The sign of calculated air mass flux is reversed so that positive values represent upward fluxes, negative values represent downward fluxes."

12) *And, I would just refer to it as the downward mass flux (rather than vertical airmass flux).*
It would be confusing to replace 'vertical mass flux" with "downward mass flux'.  In the plots, positive values are upward fluxes, negative values are downward fluxes. I used google scholar to search 'vertical air mass flux' and 'downward air mass flux'. The first term gave back more results than the second term.

13) *And, this is another confusing statement " The jet locations, approximated by the strongest winds". Jet locations are where the strongest winds are, they're not approximated. The rest of this paragraph (centered around line 315) needs to be rewritten. Make sure that all of the co-authors concur with these descriptions (Several of the paper's co-authors should be able to help make this much more understandable. They should also be able to help with the grammar issues that perhaps will also be dealt with by a copy editor.)*

On line 348 in the track changes version of the manuscript, we have modified the text as follows: "The jet, the location of maximum winds, are indicated by red thick lines."

We took suggestions by the editor and the reviewer and rewrote this section (4.1) to make the discussion clear and precise. We have tracked all the changes.

14) *lines 358/259 states " Our analysis suggests that the IAV of wave disturbances in the westerlies likely affect the IAV of the regional distributions of prevailing wind patterns as well as the strength of STE flux. " Wave disturbances (or better, just waves) and wind are going to co-vary, and the location where tropopause folds occur will co vary with the waves too. This whole discussion needs to be rewritten.*

On line 417 in the track changes version of the manuscript, we have rewritten the discussion as follows: "Our analysis suggests that significant interannual variations exist in both the regional wind patterns associated with the Westerlies waves and the strength of downward air mass fluxes across the tropopause. The IAV of stratospheric ozone influence in the troposphere reflects a combined effect of the IAV in the 3-d dynamics as well as that in the lower stratospheric ozone concentration, which may either oppose or reinforce each other."

*15) The overall results of this study seem to be*
*a) from the first part, the model seems to do OK at reproducing means and variability as based on comparisons with SBUV and sondes (and this should be an appendix or supplement.)*
*b) temporal variability in tropospheric ozone in the NH mid latitudes is related to temporal variability in the transport of ozone from the stratosphere*
*c) there is a difference in the degree of influence of stratospheric ozone on tropospheric ozone between North America and Europe. This is essentially a function of the spatial distribution of regions of tropopause folding type events, which is really a consequence of the different synoptic patterns the occur over Europe and North America.*
*None of this is really new information, but the particular analysis they do with the stratospheric tracer is fine to publish. They acknowledge a significant number of past similar studies that were done with global scale models. Although they do reference the Lin et al. CalNex paper (however, they do need to fix the citation, the paper was published several years ago), they do not acknowledge that the western US air quality issue related to STE has been studied extensively with aircraft and ground fieldwork, and I would encourage the authors to take a look at some of the papers from the LVOS experiment led by Andrew Langford.*

The two papers by Langford et al (2012;2015) have been added into references for deep STE discussion on line 61 and Line 237 in the track changes version of the manuscript.

*16) General impression; The authors need to reread all the original reviewer's comments, and respond to those comments in a more thorough manner.*

We believe that in the revised manuscript we have addressed all the comments by the reviewers.

**Reviewer 2:**

*I would like to thank the authors for taking my comments into account and revising the manuscript accordingly. However, one of my comments did not came across well. I did not mean that the model evaluation is unnecessary. My point was that the weighting how the results were described was not done in the correct way. There was to much focus on the evaluation instead of the investigation of the IAV which was supposed to be the main focus of the paper. However, with the revisions done the manuscript reads now much better and I do not feel any mismatch any longer.*

Per the request of the reviewer 1, we have deleted this section and moved the results of model evaluation from satellite observations into the supplement.

*Nevertheless, I have several technical comments that should be considered before publication:*

*P2, L45: "replay simulations" → I would suggest to either skip here "replay" or you explain what is meant with replay simulations. Alternatively, you could also add the reference to the Orbe et al. 2017 paper if you do not want to skip replay.*

We have deleted "replay" and added "full chemistry" on line 51 in the track changes version of the manuscript. We have made a few minor changes through the paragraph to clarify the description.

*P2, L63: I would suggest to skip "tropospheric" since this is confusing. Usually, CTMs simulate the entire atmosphere. If you want to emphasize that the simulation run had a focus on the troposphere it should be said more clearly.*

On line 70 in the track changes version of the manuscript, we have deleted 'tropospheric" as suggested by the reviewer.

*P2, L73: Plural or singular? Thus, either "used a nudged CCM simulation" or " used nudged CCM simulations".*

On line 80 in the track changes version of the manuscript, we have modified the text as suggested to "used a nudged CCM simulation with the ERA-Interim reanalysis…". We also revised the discussion of this citation in the same paragraph to clarify the description.

*P3, L101: Sentence "More detailed information on replay methodology can be found in Orbe et al. (2917)" is obsolete since you refer to the replay simulation and the Orbe et al. reference already a few sentences before.*

On line 109 in the track changes version of the manuscript, we deleted this sentence as suggested.

*P4, L127: in UTLS → in the UTLS*
On line 140 in the track changes version of the manuscript, the text has been modified as suggested: "in the UTLS"

*P4, L150: Centers → Center*
*P4, L159: Model did not → The model did not*
The section containing above two corrections has been deleted in the revised manuscript.

*P5, L168: relatively weaker → I would suggest to either write "relatively weak" or "weaker".*
On line 182 in the track changes version of the manuscript, we have modified the sentence as follows: "the relative contribution of stratospheric ozone is low due to…".

*P5, L171: observed ozone → specify which observations*
On line 186 in the track changes version of the manuscript, the text has been modified as follows: "ozonesonde measured ozone"

*P5, L183: lost parentheses at the end of the sentence.*
We have added the period on line 202 in the track changes version of the manuscript.

*P5, L203: …..IAV is mall, only become significant…..→ sentence not clear, please rephrase*

On line 220 in the track changes version of the manuscript, we have modified the text as follows: "The magnitude of ozone anomalies doesn't show significant seasonal difference between DJF and MAM, except over North America at 400 hPa (Table S2)."
We have also made a few minor changes through this paragraph to clarify the description.

*P7, L275: by Browell et al. (Browell et al., 1996) → by Browell et al. (1996)*

On line 301 in the track changes version of the manuscript, the reference has been modified as suggested.

*P9, L350: shows the → shows a*
On line 401 in the track changes version of the manuscript, we have changed text as suggested: "shows a".

*P9, L355: weakend descending air → please rephrase, e. g. to "weak descending air"*
On line 409-410 in the track changes version of the manuscript, we have changed the phrase into "decreased downward air mass fluxes". We rewrote the sentence as follows: "Around the tropopause, there were increased upward air mass fluxes along the west coast of North America and decreased downward air mass fluxes over western and central North America."

*P10, L376: N. America → North America*
 We have replaced all the "N. America" with "North America" in the revised manuscript.

*P10, L394: near surface → near the surface*
On line 456 in the track changes version of the manuscript, we have modified the text as suggested: "near the surface".

*P10, L398: show a similar → shows a similar*
On line 460 in the track changes version of the manuscript, the sentence containing this part has been deleted.

*P10, L403: tropospheric O3 in the troposphere → repetition, please rephrase*
 On line 465 in the track changes version of the manuscript, we have deleted "in the troposphere".

*P15, Table2: Ny Aleasund → Ny Alesund*
We have corrected the name of Ny Alesund in table 2.

*P16, Figure 1: Ny Aleasund → Ny Alesund*
 We have corrected the name of Ny Alesund in Figure 1.

*P18, Figure 4 and Figure 5: Figures should be done in a consistent way. Here, the x-axis labeling is missing (add year) and the headers should be written in the same way (same font, same abbreviation for North America).*
*P19, Figure 6: Same here as for Figures 4 and 5. Additionally, remove the y-axis labeling in the middle of the figures. N. Am should be N. American toe be consistent and height of the right 4*

*figures should be adjusted to the left 4 figures.*
*P19, Figure 6 caption: top → Top, bottom → Bottom*

Those three figures (Figure 2-4 in the revised manuscript) have been redone in a consistent way as suggested by the reviewer. The caption has been modified as suggested.

*P20 and 21, Figures 7 and 8: Increase space between the panels. What is the additional colour bar (light pink) for? Cannot this be skipped?*

These two figures (Figure 5-6 in the revised manuscript) have been redone as suggested by the reviewer. The additional color bar was generated automatically to represent the color of missing values. Since there is no missing value for model output, we have eliminated them in the revised figure.

*P20, Figure 7 caption: Do not start sentence with a number → sentence should be rephrased.*

On line 774 in the track changes version of the manuscript, we have modified the caption as suggested: " 
[revised manuscript text omitted]

---

## Author Response (AR5)

**4th Response to reviews on "Stratospheric impact on the Northern Hemisphere winter and spring ozone interannual variability in the troposphere"**

**by Junhua Liu et al.**

We truly appreciate the Editor's specific comments and suggestions. We have addressed the two points raised by the Editor.

Editor's comments:
*in line 85, please specify that replay is a SD simulation. see also comment by rev.#2 on this.*

Replay simulation is similar but not identical to a specified-dynamics simulation, since it is only driven by part of the MERRA-2 meteorology (U, V, T, pressure).

On line 79 in the track changes version of the manuscript, we have modified the text following the suggestion by the editor and the reviewer 2: "In this study, we use a long-term full chemistry GEOS-CCM replay simulation, driven by the essential output of the MERRA-2 meteorology (U, V, T, pressure), with a stratospheric ozone tracer at a horizontal resolution of 0.5°."

*rev.1 point 5: please give a brief statement (which can be based on Prather et al.) as to how the E90 tropopause compares to other tropopauses.*

On line 130 in the track changes version of the manuscript, we have added a statement of how the e90 tropopause compares to other tropopauses: "
[revised manuscript text omitted]